

# EFT and the SUSY index on the 2nd sheet

**Davide Cassani[1] and Zohar Komargodski[2]**

**1** INFN, Sezione di Padova, Via Marzolo 8, 35131 Padova, Italy
**2** Simons Center for Geometry and Physics, Stony Brook University,
Stony Brook, NY 11794-3636, USA

## Abstract

The counting of BPS states in four-dimensional $\mathcal{N} = 1$ theories has attracted a lot of attention in recent years. For superconformal theories, these states are in one-to-one correspondence with local operators in various short representations. The generating function for this counting problem has branch cuts and hence several Cardy-like limits, which are analogous to high-temperature limits. Particularly interesting is the second sheet, which has been shown to capture the microstates and phases of supersymmetric black holes in AdS$_5$. Here we present a 3d Effective Field Theory (EFT) approach to the high-temperature limit on the second sheet. We use the EFT to derive the behavior of the index at orders $\beta^{-2}, \beta^{-1}, \beta^0$. We also make a conjecture for $O(\beta)$, where we argue that the expansion truncates up to exponentially small corrections. An important point is the existence of vector multiplet zero modes, unaccompanied by massless matter fields. The runaway of Affleck-Harvey-Witten is however avoided by a non-perturbative confinement mechanism. This confinement mechanism guarantees that our results are robust.

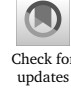

# 1   Introduction and summary

It is by now a standard fact that every 4d $\mathcal{N} = 1$ theory with a continuous $R$-symmetry can be studied on the spatial manifold $S^3$ while preserving four (time-independent) supercharges [1–4], see also the reviews [5,6]. The virtue of having the theory in compact space is that the spectrum is discrete and the states are easier to count. One can define the Hamiltonian by picking a supercharge $\mathcal{Q}$ and declaring the Hamiltonian to be proportional to $\{\mathcal{Q}, \mathcal{Q}^\dagger\}$. It is particularly interesting to consider the states which have zero energy. Those, as usual, need to be annihilated by $\mathcal{Q}, \mathcal{Q}^\dagger$.

If the original theory has $\mathcal{N} = 1$ superconformal symmetry then these zero-energy states are in one-to-one correspondence with quarter-BPS local operators. It is therefore not surprising that typically there is an infinite number of such zero-energy states and hence it is necessary to count them more carefully. This is achieved by introducing two chemical potentials $\omega_1, \omega_2$ which are conjugate to charges that commute with $\mathcal{Q}, \mathcal{Q}^\dagger$.

In this way, one is led to the refined Witten index

$$\mathcal{I}(\omega_1, \omega_2) = \text{Tr}\,(-1)^F\,e^{-\{\mathcal{Q},\mathcal{Q}^\dagger\}-\omega_1(J_1+\frac{1}{2}R)-\omega_2(J_2+\frac{1}{2}R)}\,, \tag{1.1}$$

which famously receives only contributions from states which are annihilated by $\mathcal{Q}, \mathcal{Q}^\dagger$ [7]. $R$ stands for the $R$-charge and $J_{1,2}$ are the spins corresponding to the two diagonal combinations of the Cartan generators of $SU(2) \times SU(2)$. For fermionic fields both $J_1, J_2$ must be half-integral and for bosonic fields both must be integral. Further, it is natural to view $\omega_1, \omega_2$ as independent complex parameters. (In the literature, the notation $p = e^{-\omega_1}, q = e^{-\omega_2}$ is also common.) Of course we could have put any coefficient in front of $\{\mathcal{Q}, \mathcal{Q}^\dagger\}$ in (1.1) as the index receives only contributions from states which are annihilated by $\mathcal{Q}, \mathcal{Q}^\dagger$.

There is a simple path integral interpretation for (1.1): it is computed by a Euclidean path integral on an $S^1 \times S^3$ topology with complex structure parameters $\omega_1, \omega_2$ and periodic boundary conditions for the fermion fields around $S^1$ [8–10]. The precise way the field theory is coupled to the background is determined by new-minimal background supergravity [4,11]. The metric on $S^1 \times S^3$ does not matter – it turns out to be $\mathcal{Q}$-exact (as long as it is Hermitian) [8–10]. Due to this metric-independence, we can view this construction as a holomorphic twist.[1]

Above we have identified the chemical potentials $\omega_1, \omega_2$ with the complex structure parameters on $S^1 \times S^3$. This is however only locally correct. Indeed, inspecting (1.1) it is obvious that the index is generally not periodic under $2\pi i$ shifts of $\omega_1$ or $\omega_2$. It is therefore more appropriate to think about (1.1) as a function living on a multiple cover of the space of complex structures. We can switch between the different branches by shifting $\omega_1$ or $\omega_2$ by $2\pi i$.[2]

---

[1]There is potentially a certain holomorphic anomaly [12], and see also [13–17]. This is expected to (perhaps) affect the partition function only at order $O(\beta)$ in the high-temperature expansion and hence we do not expect that it will affect this paper, apart possibly for the $O(\beta)$ term.

[2]This is reminiscent of non-modular invariant partition functions of 2d CFTs. For instance, one can think of chiral torus characters. Such objects may not be invariant under $\tau \to \tau + 1$, though this is an equivalent two-torus as far as the complex structure goes. Here the situation is similar to chiral characters: we have a holomorphic

In new-minimal supergravity, this multi-valuedness arises because it is necessary to choose an $R$-symmetry gauge field, and if the $R$-charges of various fields are not integral, certain large gauge transformations are not allowed. For instance, in $\mathcal{N} = 4$ super Yang-Mills theory, all bosonic elementary fields have $R$-charges of the form $\frac{2\mathbb{Z}}{3}$, all the fermionic fields have $R$-charges of the form $\frac{2\mathbb{Z}+1}{3}$. As a result, shifting $\omega_1 \to \omega_1 \pm 6\pi i$ or $\omega_2 \to \omega_2 \pm 6\pi i$ takes us back to the original index. In addition if we perform $\omega_1 \to \omega_1 \pm 4\pi i$ along with $\omega_2 \to \omega_2 \mp 4\pi i$ then we likewise return to the same index (this holds more generally than in $\mathcal{N} = 4$ super Yang-Mills theory). We can use this freedom to put Im $\omega_1 \in [0, 6\pi)$ and Im $\omega_2 \in [0, 2\pi)$. Hence, the index of $\mathcal{N} = 4$ super Yang-Mills theory takes values on a triple cover of the space of complex structures.[3]

A very natural problem is to try and understand the limit of small chemical potentials

$$\omega_1, \omega_2 \to 0 \tag{1.2}$$

(with $\omega_1/\omega_2$ fixed) which is in some sense a measure of the asymptotic growth of BPS operators. In the microcanonical ensemble, this corresponds to a limit of large charges $J_1 + \frac{1}{2}R$, $J_2 + \frac{1}{2}R$. It is analogous to the high-temperature/large-charge limit of ordinary statistical sums.

In 2d, for critical systems, Cardy famously argued that the high-temperature asymptotics is controlled by the trace anomaly $c$ [18]. We will see that the problem of counting heavy BPS operators is similarly controlled by central charges.

For real $\omega_1, \omega_2$, we may write $\omega_1 = \beta b/\ell$, $\omega_2 = \beta/(\ell b)$, where $\beta$ is the circumference of $S^1$ (the "thermal" circle), $\ell$ is the radius of $S^3$, and $b$ is a squashing parameter of $S^3$. Then the limit (1.2) of small complex structure parameters corresponds to the small-circle limit $\beta/\ell \to 0$ while keeping the squashing $b$ fixed. This justifies thinking about this limit as a high-temperature limit. As we will discuss in detail later, general complex values of $\omega_1, \omega_2$ can be obtained by considering a twisting of $S^3$ over $S^1$. (This is analogous to turning on $\tau_1$ in 2d theories, where it represents a twisting of the space-like circle over the thermal circle.) Note that complex values of $\omega_{1,2}$ still admit a perfectly Hermitian metric and they correspond to well-defined points in the space of complex structures. The imaginary parts of $\omega_{1,2}$ are merely certain "rotation" parameters describing the twisting of $S^3$ over $S^1$.

The problem can be therefore attacked by dimensionally reducing on the $S^1$, remembering that there could be various non-decoupling effects due to the Matsubara modes. This is a standard high-temperature expansion approach to thermal field theory, see e.g. [19]. Here, since, as we said, the fields all obey periodic boundary conditions, the KK zero modes must be treated very carefully. There are zero modes both from the matter fields as well as from the vector multiplet. The zero modes enjoy some complicated 3d dynamics. If the zero modes settle in some "nice" 3d SCFT on $\mathbb{R}^3$, one can use powerful 3d effective field theory techniques to conclude that [20]

$$\log \mathcal{I} = -\frac{8\pi^2}{3} \frac{\omega_1 + \omega_2}{\omega_1 \omega_2} (a - c) + O(1) \tag{1.3}$$

on the first sheet of the index. The asymptotics is therefore controlled by a combination of the $a, c$ trace anomalies (more generally, for non-conformal theories $a - c$ is replaced by Tr$R$, which is the 't Hooft mixed gravitational anomaly of the $R$-symmetry). Since $\log \mathcal{I}$ scales as $\sim 1/\beta$,

---

function of the complex structure parameters $\omega_{1,2}$ which is not necessarily single valued on the space of complex structures.

[3]In fact, two of the branches are related by complex conjugation. To see that pick Im $\omega_1 \in [2\pi, 4\pi)$ and Im $\omega_2 \in [0, 2\pi)$ then transform to Im $\omega_1 \in [-2\pi, 0)$ and Im $\omega_2 \in [4\pi, 6\pi)$ and then reversing the signs of both Im $\omega_{1,2}$, which corresponds to complex conjugation of the index, we find Im $\omega_1 \in (0, 2\pi]$ and Im $\omega_2 \in (-6\pi, -4\pi]$ and this can be finally transformed to Im $\omega_1 \in (0, 2\pi]$ and Im $\omega_2 \in (0, 2\pi]$. Therefore, we have come back to where we started. In particular the limit Im $\omega_1 \to 2\pi^+$ together with Im $\omega_2 \to 0^+$ is identified with the limit Im $\omega_{1,2} \to 2\pi^-$ upon complex conjugation, which is tantamount to the statement that the Cardy limit on the 3rd sheet is the complex conjugate of the Cardy limit on the 2nd sheet.

which is like one positive power of the temperature, this is reminiscent of a 2d growth of operators. It is tantalizing that a similar slow growth is observed in non-supersymmetric theories with periodic boundary conditions for the fermions [21]. The $O(1)$ corrections to (1.3) are not universal and in particular receive a model-dependent contribution from the $f$-coefficient of the SCFT in 3d. By contrast, below we will study a different limit where the $O(1)$ correction is universal (on any suitable three-manifold).

A key assumption in the discussion above was that the 3d zero modes settle in a "nice" 3d SCFT. Let us now explain what "nice" means in this context. In Lagrangian theories, upon studying the theory on a finite $S^1$, the holonomies of the vector multiplet gauge fields become periodic scalar fields $u$ which are always flat at tree level and need to be integrated over. In some cases, they may remain flat to all orders in perturbation theory on $S^1 \times \mathbb{R}^3$. Typically a potential is induced on $S^1 \times S^3$, $V^{\text{eff}}(u)$, and since the potential has terms scaling with negative powers of $\beta$ one has to simply account for the saddle points of $V^{\text{eff}}(u)$ in order to understand the behavior of the theory at $\beta \to 0$. The theory is "nice" if $u = 0$ is a saddle point and it is the dominant one (one can relax this a little, allowing the moduli space of $u$ to not be lifted at all). This condition seems to hold quite generally. In fact, for $a \leq c$ and for charge conjugation invariant theories (vector-like theories) it appears to be always the case, as far as we know [22–25]. It seems reasonable to conjecture that (1.3) indeed holds for charge-conjugation invariant theories with $a \leq c$, even for non-Lagrangian theories. Since $\mathcal{N} = 2$ theories are always vector like, this might explain why indeed (1.3) always seems to hold when $a \leq c$ [26].

Having emphasized that the index is in fact defined on a multiple cover of the space of complex structures, it becomes evident that (1.2) is not the only possible "high-temperature" limit. For instance, we can first go to the second sheet and only then take the $S^1$ to shrink.[4]

Recently, starting with [27–29], prompted by [30], there has been a lot of activity on the holographic microstate counting for supersymmetric black holes in $\text{AdS}_5$. The analysis of the Cardy-like limit on the second sheet has played an important role in displaying the correct asymptotic growth accounting for the black hole microstates, see [28, 31–38]. This is complementary to the study of the large-$N$ limit of the index, discussed in [29, 39–47]. If one takes the large-$N$ limit first, there are many saddles (some of which have a direct holographic interpretation within the classical gravitational theory). If one subsequently takes the Cardy-like limit on the second sheet, a drastic simplification occurs and a clear dominating saddle, corresponding to the black hole, is singled out.[5] If one takes the limits in the opposite order, i.e. implementing the large-$N$ limit after taking the Cardy-like limit on the second sheet, the large $N$ limit becomes straightforward. The Cardy limit on the second sheet is therefore quite an interesting object to study and it is for this reason that we dedicate the paper to the physics on the second sheet.

To land on the second sheet we transform $\omega_1 \to \omega_1 + 2\pi i$ and $\omega_2 \to \omega_2$ in terms of which we can rewrite the index (1.1) as [27, 33, 34]

$$\mathcal{I} = \text{Tr}\, e^{-\pi i R}\, e^{-\{\mathcal{Q}, \mathcal{Q}^\dagger\} - \omega_1 \left(J_1 + \frac{1}{2}R\right) - \omega_2 \left(J_2 + \frac{1}{2}R\right)}. \tag{1.4}$$

One can readily see that this index is again sensitive only to ground states since for massive representations the phase $e^{-\pi i R}$ leads to exact cancelations multiplet-by-multiplet; because of this weighting by $e^{-\pi i R}$, we may refer to (1.4) as the *R-charge index*. It is interesting that among the ground states, the phase $e^{-\pi i R}$ leads to somewhat less cancelations than the previous $(-1)^F$ and this is why on the second sheet indeed the index $\mathcal{I}$ typically grows faster in the new Cardy-

---

[4]In 2d CFTs, going to the second sheet is achieved by acting with the $T$ matrix – see Footnote 2.

[5]Numerical analyses show hints of the gravitational behavior of the index already at finite (moderately large) $N$ [48, 49].

like limit:

$$\omega_1, \omega_2 \to 0 \,. \tag{1.5}$$

One can give a direct path integral interpretation for the index on the second sheet (1.4). It is computed by a Euclidean path integral on an $S^1 \times S^3$ topology with complex structure parameters $\omega_1, \omega_2$ as before, however now we are imposing different boundary conditions for the fields around $S^1$. Indeed, since in (1.4) the $(-1)^F$ insertion is replaced by $e^{-\pi iR}$, we have that for any field $\chi$ with $R$-charge $R$ and fermion number $F$, when we go once around $S^1$ we get the twisted identification

$$\chi(\tau + \beta) = e^{\pi i(R+F)}\chi(\tau) \,, \tag{1.6}$$

$\tau$ being the coordinate on $S^1$.[6] Unlike the situation on the first sheet, now the chiral multiplet fields generically have no zero modes on the circle since generically $R \neq 0$ mod 2 for chiral multiplets. On the other hand, the vector multiplet fields have zero modes since the $R$-charge of the gaugino is always 1. It is also important to notice that since the supercharges commute with the $R + F$ operator defining the boundary conditions, a reduction along $S^1$ preserves supersymmetry.[7]

At tree-level, the low-energy theory after the circle reduction is therefore a pure 3d $\mathcal{N} = 2$ vector multiplet. This model famously [50] has a runaway behavior and no stable vacuum. It might therefore appear that the small circle limit (1.5) is problematic.

An important property of the second sheet is that because of the phase (1.6) the KK modes are in fact generically not symmetric about the origin (we will sometimes say that they are not "vector like"). For this reason, the non-decoupling effects of the KK modes could influence the non-perturbative dynamics of the 3d $\mathcal{N} = 2$ vector multiplet! Indeed, a Chern-Simons term for the vector multiplet is induced and it so happens that this Chern-Simons term is just right to guarantee that in the limit (1.5) the vector multiplet theory flows to a trivial, confined vacuum. For $SU(N)$ gauge theory, the Chern-Simons term is at level $N$. This also explains the recent observation of [37] that an $SU(N)$ Chern-Simons theory at level $N$ emerges at $O(\beta^0)$ from the counting problem in certain gauge theories in four dimensions. The situation outlined above is ideal in order to develop a consistent EFT in the "high-temperature" limit. At order $O(\beta^0)$ in the expansion there will be a contribution from the gapped degrees of freedom of the zero modes, while at orders $O(\beta^{-2}), O(\beta^{-1})$ the zero modes do not contribute by virtue of them being gapped in a healthy vacuum. Since the zero modes are gapped, one can equally easily make predictions for the behavior of the index on a large class of spatial three-manifolds.

In terms of the holonomy potential $V^{\text{eff}}(u)$, the claims above can be re-stated by saying that on the second sheet the origin $u = 0$ is always a saddle in a certain range of the chemical potentials. The potential grows like $V^{\text{eff}}(u) \sim u^2/\beta^2$ around the origin which in the effective field theory on $S^1 \times \mathbb{R}^3$ we interpret by saying that the Chern-Simons term lifts the Coulomb branch. This renders the EFT approach more robust on the second sheet than on the first sheet, where the existence of a minimum at the origin relies on some additional assumptions which we reviewed above. Due to the universal nature of the second sheet, where the Coulomb branch around the origin is lifted by a level $N$ Chern-Simons term, the predictions of the corresponding EFT are more universal.

In fact we will see an example of a theory which is not charge conjugation invariant and has $c > a$, where the two sheets are identical in terms of the microscopic counting problem but look very different in terms of the $S^1$ reduction. There is complicated dynamics on the first

---

[6]More generally the shift $\omega_1 \to \omega_1 + 2\pi i n_0$, with $n_0$ an integer, leads us to explore the different sheets of the refined index, whenever they exist, and gives the twisted identifications $\chi(\tau + \beta) = e^{\pi i n_0(R+F)}\chi(\tau)$.

[7]An effective field theory approach to this problem was previously discussed in [28, 33]. Here we present a manifestly supersymmetric reduction and the metrics both in 4d and in 3d are regular and real.

sheet with a nontrivial minimum for the holonomies that "conspires" to reproduce exactly the predictions of the second sheet with the minimum at the origin.

Due to the zero modes being lifted on the second sheet, their contribution and that of the massive KK modes are captured by 3d local contact terms. Similarly to the analysis of [20] on the first sheet, the coefficients of these contact terms are regularized sums over all Matsubara frequencies (albeit with different charges under the KK gauge symmetry due to our twisted identifications (1.6)).

Putting the contribution of all contact terms together, the result we obtain is

$$
\log \mathcal{I} = \frac{1}{48\omega_1\omega_2}\Big[ -8\pi^3 i\,(\mathrm{Tr}R^3 - \mathrm{Tr}R) - 4\pi^2(\omega_1 + \omega_2)(3\,\mathrm{Tr}R^3 - \mathrm{Tr}R)
$$
$$
+ 6\pi i\,(\omega_1 + \omega_2)^2\,\mathrm{Tr}R^3 - 2\pi i\,(\omega_1^2 + \omega_2^2)\,\mathrm{Tr}R\Big] + \log|G| + O(\beta). \tag{1.7}
$$

$\mathrm{Tr}R^3$ and $\mathrm{Tr}R$ are the $R$-symmetry 't Hooft anomalies and they can be rewritten in terms of $a, c$ if the $R$-symmetry is the superconformal one. The meaning and origin of $\log|G|$ will be explained below. This coincides with the $O(\beta^{-2})$ term obtained in [33,34], the $O(\beta^{-1})$ term given in [34] as well as the $O(\beta^0)$ term recently found in a class of theories for $\omega_1 = \omega_2$ [37].

We leave to future work the detailed clarification of the $O(\beta)$ terms. This problem is well defined since at $O(\beta)$ there is no covariant and gauge-invariant supersymmetric counter-term in 4d [51,52] (see [14] for a discussion of $O(\beta)$ terms in relation with the matter of Footnote 1). It has been argued in [37] that all terms beyond $O(\beta)$ are exponentially suppressed. Let us give an argument to that effect from our EFT approach. Since we are evaluating a supersymmetric effective action on a background that preserves two supercharges with opposite $R$-charge, local terms that are true $D$-terms must evaluate to zero. The Chern-Simons terms contributing to (1.7) are an exception since they are not given by a supersymmetry variation of well defined (gauge invariant) quantities. Since one does not expect any Chern-Simons-like terms beyond those we investigate here, it then must be true that all the other covariant local terms are true $D$-terms and the high-temperature expansion for the partition function $Z$ truncates, up to exponentially small terms. For instance, the curvature-squared invariants given in [53] are of this type. (It would be nice to be able to understand the exponentially small terms in terms of an EFT language – see [54] and references therein for a study of exponentially small corrections in a different EFT.) These arguments imply that $\log Z$ should already truncate at $O(\beta^0)$. However, here one must remember that the index and partition function differ by the supersymmetric Casimir energy [9,52,55] at $O(\beta)$, $\log Z = -\beta E_{\mathrm{Casimir}} + \log \mathcal{I}$, and hence the small-$\beta$ asymptotics of $\log \mathcal{I}$ must contain precisely the Casimir energy at $O(\beta)$ (see [32] for a related discussion). This seems perfectly consistent with the examples we considered. Therefore, if the above statement about $D$-terms can be turned into a proof, one can then easily extend the predictions of the EFT to all perturbative orders in $\beta$ by including $\beta E_{\mathrm{Casimir}} = \frac{(\omega_1+\omega_2)^3}{48\omega_1\omega_2}(\mathrm{Tr}R^3 - \mathrm{Tr}R) + \frac{\omega_1+\omega_2}{24}\mathrm{Tr}R$ into (1.7)[8] and rearranging the expression so as to obtain:

$$
\log \mathcal{I} = \frac{(\omega_1 + \omega_2 + 2\pi i)^3}{48\,\omega_1\omega_2}\mathrm{Tr}R^3 - \frac{(\omega_1 + \omega_2 + 2\pi i)(\omega_1^2 + \omega_2^2 - 4\pi^2)}{48\omega_1\omega_2}\mathrm{Tr}R
$$
$$
+ \log|G| + O(e^{-\ell/\beta}). \tag{1.8}
$$

This exhausts all the analytic terms in the Cardy limit on the second sheet.[9] For large-$N$ holographic theories, where the $\mathrm{Tr}R^3$ term dominates, (1.8) agrees with the function proposed

---

[8]Here we are using the prefactor $\beta E_{\mathrm{Casimir}}$ given in [52], analytically continued to complex chemical potentials. The validity of this continuation has been demonstrated in [56] for the twisted background of interest to us. The prefactor of [27] was computed using a slightly different background gauge field.

[9]Continuing with our analogies to 2d, it is true that the expansion of $\log Z$ at finite temperature in any 2d

in [30] and derived as a supergravity on-shell action in [27], whose Legendre transform gives the black hole entropy.

Let us discuss some important issues concerning the regime of validity of (1.7) (and conjecturally (1.8)).

- The most singular piece in (1.7) is $-\frac{\pi^3 i}{6\omega_1 \omega_2}\left(\text{Tr}R^3 - \text{Tr}R\right)$. This corresponds to a purely oscillating behavior of the index (1.4) for real $\omega_{1,2} \to 0$. Therefore one needs to have some small twisting of the $S^3$ over the $S^1$ to find the desirable exponentially growing density of operators. For instance we may take $\omega_1 \to 0^+$ and $\omega_2 = (1-i)\omega_1$. Then, if $R$ above corresponds to the superconformal $R$-symmetry, we would always find an exponentially growing index on the second sheet since $\text{Tr}R^3 - \text{Tr}R \geq 0$ is guaranteed to hold by unitarity [57].[10] A similar small imaginary piece in $\omega_{1,2}$ is necessary to make our discussion of the $u = 0$ minimum of $V^{\text{eff}}(u)$ rigorous.

- The effective field theory techniques we use allow us to establish that there is a local minimum of $V^{\text{eff}}(u)$ at $u = 0$ and they allow us to make predictions for the corresponding contributions to the index at each order in $\beta$. These facts are model independent and presumably hold also for non-Lagrangian theories. It is however not possible to use the effective field theory techniques discussed here to say something general about the possibility of other minima of $V^{\text{eff}}(u)$, away from the origin. One example where other degenerate minima must exist is when we have a one-form symmetry in the original 4d theory, e.g. $\mathbb{Z}_N$ in $\mathcal{N} = 4$ theory with gauge group $SU(N)$. Since $u = 0$ breaks it spontaneously we ought to have additional $N-1$ exactly degenerate local minima, each contributing in the same way to the index. This is the standard situation in theories with a spontaneously broken one-form symmetry [59]. Therefore, in any theory with a one-form symmetry (Abelian) group $G$ of order $|G|$, there will be a $\log|G|$ term in the asymptotics of the index as in (1.7). It is tempting to hypothesize that no other minima can exist save for those guaranteed by the spontaneously broken one-form symmetry (if a one-form symmetry is present). It would be interesting to investigate this question in the future. Needless to say, the existence of minima that are more dominant than the minimum at the origin would invalidate (1.7) and (1.8). No such example is presently known.

We can also consider more general backgrounds involving a three-manifold $\mathcal{M}_3$ different from the squashed three-sphere discussed so far. For instance, we can take $\mathcal{M}_3$ to be a Lens space (or a more general Seifert manifold) compatible with the $R$-charge assignments, twist it over $S^1$, and arrange the new-minimal supergravity auxiliary fields so as to obtain a supersymmetric background. The corresponding supersymmetric index (see [60, 61]) in general should admit a second sheet, and our EFT analysis should still capture some of the relevant physics. Indeed, since the zero modes are trivially gapped, all we need to do is to recompute the integrals (2.4)–(2.7) below, and this will automatically provide a prediction for the $\beta^{-2}, \beta^{-1}, \beta^0$ terms. This is conceptually different from the first sheet, where there is typically a massless theory at the origin and hence the $\beta^0$ term in the asymptotic expansion has to be studied on a case-by-case basis. The prediction for the $\beta^{-2}, \beta^{-1}, \beta^0$ terms on the second sheet should be thus completely universal by virtue of the tree-level zero modes flowing to a gapped theory at the origin. To adapt the prediction of the $O(\beta)$ term to Seifert manifolds one may

---

critical system truncates up to exponentially small terms with the local term which is extensive in the volume of space, i.e. $\ell/\beta$, where the coefficient is famously proportional to the central charge and $\ell$ is the length of the spatial slice. This is simply because no local term other than $\int dx \sqrt{g}$ can be written in one space dimension. The leading exponentially small correction is determined by the scaling dimensions of the first nontrivial operator.

[10]The case $\text{Tr}R^3 - \text{Tr}R = 0$ is presumably never realized in interacting theories [58].

use the supersymmetric Casimir energy given in [61]. (Here we are talking, as before, about the contribution from the universal saddle at the origin.)

Another possible extension of our approach is to 6d theories, which should similarly have a second sheet and a manifestly supersymmetric 5d effective theory describing the high-temperature expansion. See [28,62–64] for some work on the subject, building on the method developed for the first sheet [20,65,66].

The rest of the paper is organized as follows. In Section 2 we discuss some general features of the small-circle expansion of the 4d partition function and introduce the relevant supersymmetric local terms. In Section 3 we study the zero modes and show that they are governed by a Chern-Simons gauge theory. In Section 4 we combine the zero mode contribution with the contribution arising from integrating out the massive modes. We give the result for the $\beta \to 0$ limit of the partition function up to $O(\beta^0)$ for the case of real parameters $\omega_1, \omega_2$ and extend it to complex values by analyticity. In Section 5 we study some examples, including a chiral theory. In Section 6 we evaluate the supersymmetric contact terms in a twisted background with general complex structure parameters $\omega_1, \omega_2$ and prove that the analytic continuation done in the previous section is correct (modulo one integral that we had some difficulties with). Some details about the KK reduction of the 4d supergravity multiplet are collected in an appendix.

As this paper was being completed, the preprint [67] discussing related topics appeared.

## 2 General features of the 3d effective action

We start our discussion by recalling some general features of the supersymmetric 3d effective field theory describing the asymptotics of the 4d index. In particular, we introduce the relevant contact terms.

Let us assume that the four-dimensional theory is trivially gapped on $S^1 \times \mathbb{R}^3$. Then the path integral on $S^1 \times \mathcal{M}_3$ (with $\mathcal{M}_3$ much larger than the $S^1$) can be captured by local terms for background fields on $\mathcal{M}_3$. The reason is that since the theory is gapped in infinite volume, correlation functions on $S^1 \times \mathcal{M}_3$ are exponentially decaying as we take points on $\mathcal{M}_3$ far from each other. Hence the effective action is purely made of *contact terms*, that is, analytic functionals of the background fields. This gives an expansion of the free energy in powers of $\beta$ and it captures all the terms in this expansion save for the exponentially small ones. As we will show soon, our problem is exactly of this nature.

In general, for massless theories on $S^1 \times \mathbb{R}^3$, the dependence of the partition function $Z$ on the geometry of $\mathcal{M}_3$ is hopelessly complicated. But for gapped theories it admits a simple expansion in the inverse size of $S^1$ where each term in the expansion is a local integral on $\mathcal{M}_3$.

Since as we argued in the introduction the $S^1$ dimensional reduction preserves supersymmetry (even when we consider the twisted boundary conditions (1.6)), the contact terms must respect $\mathcal{N} = 2$ supersymmetry in three dimensions. In the present work the only choices of $\mathcal{M}_3$ that we will discuss are $\mathbb{R}^3$ and $S^3_b$ (but many other choices are possible, too) where $S^3_b$ is the supersymmetric squashed three-sphere background, which may also be twisted over $S^1$. The spectrum on $S^3_b$ is discrete and the volume is finite so the partition function is a very nice object to study.

In the general setting of the small circle limit $\beta \to 0$, the most straightforward terms one can imagine may contribute to $\log Z$ are $\sim \beta^{-3} \int \mathrm{d}^3 x \sqrt{g}$, $\sim \beta^{-1} \int \mathrm{d}^3 x \sqrt{g} \, \mathcal{R}$ etc., where $g$ is the metric on $\mathcal{M}_3$ and $\mathcal{R}$ is the Ricci scalar. The first term corresponds to the usual thermal free energy density and the second term gives a sub-extensive correction at finite volume. The coefficients of these terms are usually very hard to compute analytically. However in the present setting where we have $\mathcal{N} = 2$ supersymmetry the term $\beta^{-3} \int \mathrm{d}^3 x \sqrt{g}$ is forbidden since

it cannot be supersymmetrized [68].[11] The term $\beta^{-1} \int \mathrm{d}^3 x \sqrt{g}\, \mathcal{R}$ is luckily accompanied due to supersymmetry by a certain Chern-Simons term and the latter is one-loop exact. In fact, up to and including order $O(\beta^0)$, all the possible $\mathcal{N} = 2$ supersymmetric contact terms are related to various Chern-Simons terms and are one-loop exact. If the argument given in the introduction about all other possible contact terms being true $D$-terms is correct, the expansion of $\log Z$ will therefore truncate at $O(\beta^0)$, up to exponentially small terms.

A four-dimensional $\mathcal{N} = 1$ theory with an $R$-symmetry can be coupled to background new-minimal supergravity [4, 11]. In addition to the metric, new-minimal supergravity [68, 69] comprises the $R$-symmetry $U(1)$ gauge field and a (globally well-defined) one-form background field. Consequently, the gauge fields obtained after the circle reduction are the KK photon gauge field $c$ arising from the 4d metric, the $U(1)_R$-gauge field $\check{A}$, and we also have the 3d spin connection $\omega^{ab}$ (we use the $\check{\phantom{x}}$ symbol to distinguish 3d quantities from similar 4d quantities). It is convenient to take $c = c_i \mathrm{d}x^i$ to have the dimension of a length and $\check{A} = \check{A}_i \mathrm{d}x^i$ to have dimension 0. Similarly, the spin connection has dimension 0 as a one-form. Up to normalization, we can hence form four independent Chern-Simons terms

$$c \wedge \mathrm{d}c, \qquad \check{A} \wedge \mathrm{d}c, \qquad \check{A} \wedge \mathrm{d}\check{A}, \qquad \mathrm{tr}\left( \omega \wedge \mathrm{d}\omega + \frac{2}{3}\, \omega \wedge \omega \wedge \omega \right), \tag{2.1}$$

the last one being the gravitational Chern-Simons term. From dimensional analysis it is clear that $c \wedge \mathrm{d}c$ and its supersymmetric partners will lead to a contribution at order $\beta^{-2}$, $\check{A} \wedge \mathrm{d}c$ and its partners will lead to a contribution at order $\beta^{-1}$ and $\check{A} \wedge \mathrm{d}\check{A}$, the gravitational Chern-Simons term and their partners lead to contributions of order $\beta^0$. Each of these must be turned into an $\mathcal{N} = 2$ supersymmetric contact term in 3d.

The reduction of the 4d new-minimal supergravity multiplet yields the 3d new-minimal supergravity multiplet and an abelian gauge vector multiplet, that we will dub the *KK photon multiplet* as it contains the KK photon. For the structure of 3d new-minimal supergravity see e.g. [53, 70–72]. The bosonic components of the 3d new-minimal supergravity multiplet are given by

$$\text{supergravity multiplet} = \left( \check{g}_{ij}\, , \, \check{A}_i\, , \, \check{V}_i\, , \, H \right), \tag{2.2}$$

where in addition to the 3d metric $\check{g}_{ij}$ and the $R$-symmetry gauge field $\check{A}_i$, we have the globally well-defined one-form $\check{V}_i$, satisfying $\check{\nabla}_i \check{V}^i = 0$, and the scalar $H$. The KK photon multiplet has bosonic components

$$\text{KK photon multiplet} = \left( \sigma_{\mathrm{kk}}\, , \, c\, , \, D_{\mathrm{kk}} \right), \tag{2.3}$$

where $\sigma_{\mathrm{kk}}$ is a real mass scalar field and $D_{\mathrm{kk}}$ is the auxiliary field in the vector multiplet. Using these we can construct the supersymmetric completion of the background Chern-Simons terms (2.1), see Section 6.3 for some more details. The resulting terms are:

*1) KK photon Chern-Simons term*

$$I_1 = \frac{1}{4\pi} \left( \frac{2\pi}{\beta} \right)^2 \int \left( i\, c \wedge \mathrm{d}c - 2\sigma_{\mathrm{kk}} D_{\mathrm{kk}} \mathrm{vol}_3 \right), \tag{2.4}$$

*2) mixed KK photon–R-symmetry Chern-Simons term*

$$I_2 = \frac{1}{\beta} \int \left[ i\left( \check{A} - \frac{1}{2}\check{V} \right) \wedge \mathrm{d}c - H D_{\mathrm{kk}} \mathrm{vol}_3 - \frac{1}{4}\, \sigma_{\mathrm{kk}} \left( \check{\mathcal{R}} + 2\check{V}^i \check{V}_i + 2H^2 \right) \mathrm{vol}_3 \right], \tag{2.5}$$

*3) R-symmetry Chern-Simons term*

$$I_3 = \frac{1}{4\pi} \int \left[ i\left( \check{A} - \tfrac{1}{2}\check{V} \right) \wedge \mathrm{d}\left( \check{A} - \tfrac{1}{2}\check{V} \right) - \frac{1}{2} H\left( \check{\mathcal{R}} + 2\check{V}^i \check{V}_i + 2H^2 \right) \mathrm{vol}_3 \right], \tag{2.6}$$

---

[11]We thank M. Rocek for discussions on this topic.

*4) gravitational Chern-Simons term*

$$I_4 = \frac{i}{192\pi} \int \left[ \text{tr}\left( \omega \wedge d\omega + \frac{2}{3} \omega \wedge \omega \wedge \omega \right) + 4\left( \check{A} - \frac{3}{2}\check{V} \right) \wedge d\left( \check{A} - \frac{3}{2}\check{V} \right) \right], \qquad (2.7)$$

where $\check{\mathcal{R}}$ is the Ricci scalar of the 3d metric.

Another basis of contact-terms that is very slightly more convenient for computations is

$$I'_1 = I_1 , \quad I'_2 = I_2 , \quad I'_3 = I_3 , \quad I'_4 = I_4 - \frac{1}{12} I_3 . \qquad (2.8)$$

The slight computational advantage of the $I'$ basis is that each Chern-Simons term appears exactly once.

In order to obtain our effective description of the index asymptotics, we should evaluate the contact terms above in the relevant background, and determine the respective coefficients.

We can evaluate the contact terms on the supersymmetric squashed sphere $S^3_b$ background of [73], for instance. This gives [24, 72, 74]

$$I_1 = \frac{4\pi^3 i \ell^2}{\beta^2}, \quad I_2 = \frac{2\pi^2 \ell(b + b^{-1})}{\beta}, \quad I_3 = -\frac{\pi i (b + b^{-1})^2}{4}, \quad I_4 = \frac{\pi i}{48} \left[ (b - b^{-1})^2 + 2\nu \right]. \qquad (2.9)$$

When evaluating the gravitational Chern-Simons term in (2.7) we have taken into account the familiar framing anomaly. Due to the non-invariance of the spin connection under frame rotations, different choices of frame allow to shift $I_4$ by $\frac{\pi i}{24} \nu$, where $\nu$ is an integer, and thus multiply the partition function by $e^{\pi i k_g/24}$ [75], where in our normalization, $k_g = 1$ from integrating out a Dirac fermion. The choice $\nu = 0$ in the expression (2.9) is natural from the 3d point of view and indeed this arises naturally from localization: $\nu = 0$ corresponds to a frame adapted to the transversely holomorphic foliation (THF). Even if it is not necessarily fixed from the 3d point of view, from the 4d point of view it is, since there cannot be a corresponding counter-term. When the contact term is evaluated in a frame compatible with the complex structure in 4d, as in (2.9), we have $\nu = 0$. We will make this choice in the following.

The 3d $S^3_b$ background of [73] is the KK reduction of an $S^1 \times S^3_b$ direct product background where the complex structure parameters $\omega_1, \omega_2$ are real. In order to make the story complete we need to, and in fact we must, consider a more general background where $S^3_b$ is twisted over $S^1$ (generalizing the construction of [76]), so as to encode general complex structure parameters $\omega_1, \omega_2$. We will also need to develop the corresponding dictionary between the 4d supergravity fields and the 3d supergravity fields. This will allow us to express the answer using the generically complex chemical potential $\omega_1, \omega_2$ that appeared in the superconformal index. We will first sidestep this exercise and arrive at our final answer in Section 4 using holomorphy and later in Section 6 we will fill this gap.

As explained in [20], the coefficients of the contact terms $I_{1,2,3,4}$ are obtained by summing the contribution of the infinite KK modes (i.e. Matsubara frequencies), weighted by the charges of the KK modes under the KK photon and under the $R$-symmetry gauge field. The contribution of each particular KK mode is essentially as originally found by [77] (see also [72, 74] for additional details). The sum over the infinite KK tower is divergent and has to be suitably regularized.

Given the twisted boundary condition (1.6) we are considering, relevant to describe the index on the second sheet, the expansion in Fourier modes on $S^1$ for any field $\chi$ of $R$-charge $R$ is

$$\chi(\tau) = \sum_{n \in \mathbb{Z}} \chi_n \, e^{\frac{2\pi i}{\beta}\left(n + \frac{R+F}{2}\right)\tau} . \qquad (2.10)$$

Hence, the KK charge and mass of the KK mode $\chi_n$ is

$$q_n = n + \frac{R+F}{2}, \qquad m_n \sim \frac{1}{\beta}\Big(n + \frac{R+F}{2}\Big). \qquad (2.11)$$

In the following we first discuss the physics of the zero modes and then we combine their contribution with that of the rest of the KK tower.

# 3   Analysis of the zero modes

## 3.1   $SU(N)_N$ Chern-Simons dynamics

Since we are ultimately interested in the limit of small $S^1$, it is useful to set up an effective theory on $S^1 \times \mathbb{R}^3$ and then turn the $\mathbb{R}^3$ into a large $S^3$. This is the standard approach to the high-temperature limit. For that we must first and foremost understand the fate of the zero modes remaining after imposing the boundary conditions (1.6) on the $S^1$. As we said, generically, the only zero modes are in the vector multiplet. To prepare, let us first review the dynamics of the $\mathcal{N} = 2$ vector multiplet with and without a Chern-Simons term. For simplicity we will focus on the $SU(N)$ vector multiplet with level $k \in \mathbb{Z}$ Chern-Simons term.

- The $k = 0$ theory has a runaway (no stable vacuum) [50]. This is what we find when doing naive dimensional reduction on the $S^1$ before taking into account the non-decoupling effects of the KK modes.

- For $0 < |k| < N$ the theory has a SUSY breaking vacuum. The vacuum supports a Goldstino Dirac fermion and a nontrivial TFT, $U(N-k)_{k,N}$ (for $0 < k < N$ and a similar result for $-N < k < 0$) Chern-Simons theory [78]. Neither the SUSY breaking nor the nontrivial TFT can be seen in perturbation theory.

- For $|k| > N$ the theory has a gapped SUSY vacuum with the low energy theory being the $SU(N)_{k-N}$ TFT for $k > N$ and $SU(N)_{-k+N}$ TFT for $k < -N$. This can be seen by a weak coupling analysis for $|k| \gg N$ where the gaugino fermions can be integrated out at weak coupling since they are heavy. The TFTs $SU(N)_{k-N}$ or $SU(N)_{-k+N}$ lead to deconfinement and a nontrivial vacuum degeneracy of SUSY vacua on the torus for $|k| > N$.

- For $k = \pm N$ the theory confines and there is a unique gapped trivial vacuum.

It is a surprising fact that regardless of the original $\mathcal{N} = 1$ theory that we started from, we obtain $k = N$ from integrating out the KK modes. By the above classification of the phases of the vector multiplet, our zero modes are therefore trivially gapped and confined on $\mathbb{R}^3$. This is a very promising starting point for the effective theory in the small $S^1$ limit. The fact that the zero modes furnish the $k = N$ Chern-Simons theory for the vector multiplet will be crucial to explain the recent observation of [37] that in the high-temperature limit on the second sheet of $\mathcal{N} = 4$ SYM and $\mathcal{N} = 1$ quiver gauge theories, a matrix model of a supersymmetric Chern-Simons theory miraculously emerges. Our results should extend to more general gauge theories and gauge groups, thus explaining similar observations made in [38].

Let us now derive the fact that $k = N$ always holds. Consider some gauge theory with $SU(N)$ gauge group and chiral multiplets with $R$-charges $R_I$ in some representations $\mathcal{R}_I$ of the gauge group. The low-energy theory for small $S^1$ is therefore a 3d $\mathcal{N} = 2$ $SU(N)$ vector multiplet. Due to the phase $e^{\pi i R_I}$ that is picked up by chiral fermions the spectrum of KK modes is not vector-like. The masses of the various KK modes are $\frac{1}{\beta}(n + R_I/2)$ for $n \in \mathbb{Z}$. Integrating out these modes induces a CS term for the dynamical $SU(N)$ gauge fields. The condition that

the $R$-symmetry is a true symmetry (anomaly free) in the original 3+1 dimensional theory reads:

$$\sum_{I \in \text{chirals}} (R_I - 1)T(\mathcal{R}_I) + N = 0 \, . \tag{3.1}$$

On the other hand, upon a KK reduction with circle boundary conditions with a phase $e^{i\pi R_I}$ for every such chiral multiplet, the induced CS term for the dynamical gauge fields is given by $T(\mathcal{R}_I)sgn(n + R_I/2)$ from every KK mode of a chiral multiplet in representation $\mathcal{R}_I$. There is no contribution from massive KK gaugino modes since they are vector like. Therefore the coefficient of the CS term for the zero modes reads

$$k_{\text{dynamical}} = \sum_{I \in \text{chirals}} T(\mathcal{R}_I) \sum_n sgn(n + R_I/2) = \sum_{I \in \text{chirals}} T(\mathcal{R}_I)(1 - R_I) = N \, . \tag{3.2}$$

In the last step we used the anomaly-free condition (3.1) and in the step before that we used the standard result of zeta function regularization.[12]

Therefore, the tree-level zero modes flow to $SU(N)_N$ $\mathcal{N} = 2$ theory which in the deep infrared is trivially gapped and confined. It follows that the partition function of the zero modes is analytic in all external parameters and in particular cannot influence the singular terms in the high-temperature expansion. Furthermore, since we know a lot about $SU(N)_N$ $\mathcal{N} = 2$ theory from localization, we can use it to make a prediction about the $\beta^0$ term in the small $\beta$ expansion.

The argument around (3.2) easily generalizes to any quiver gauge theory, conformal or not. We get after the circle reduction a product gauge group each at the exactly correct level that leads to a trivial gapped vacuum.

While the arguments so far that led to $SU(N)_N$ $\mathcal{N} = 2$ theory were very much using a Lagrangian formalism and an explicit mode expansion, one can hope to go beyond that. Here we make a very modest remark in that direction. Starting from some abstract 4d $\mathcal{N} = 1$ theory that may not have a Lagrangian, it is believed that the circle reduction does have a Lagrangian description, see e.g. the examples of [79]. We can then ask what is the Chern-Simons level of a certain $SU(N)$ gauge theory node. At least for theories with a one-form symmetry, it is easy to prove that the level has to be an integer multiple of $N$. Otherwise, the one-form symmetry would be anomalous [80–82]. By contrast, in the original theory the one-form symmetry is non-anomalous since there cannot be a pure one-form symmetry anomaly in 4d and there cannot be a mixed anomaly with a continuous $R$-symmetry either.

There are some exceptions that need to be considered separately: one is the case where $R_I = 0, 2$ as then we have matter zero modes and the dynamics has to be re-considered. An additional subtlety arises if some of the gauge groups have $U(1)$ factors, then we find that $\sum_I (R_I - 1)T(\mathcal{R}_I) = 0$ (now $T(\mathcal{R}_I)$ is proportional to the square of the $U(1)$ charge). In that event the induced CS term for the dynamical $U(1)$ gauge fields is 0. Since $U(1)_0$ gauge theory has a flat direction this again leads one to worry about the fate of the vacuum at $\beta = 0$. However in this case monopole operators can be induced and the situation needs to be considered more carefully (along the lines of [83]). We will not discuss these two subtle cases here.

---

[12]Throughout we use the following identities:

$$\sum_{n \in \mathbb{Z}} sgn(n + \tfrac{R}{2}) = 1 - R,$$

$$\sum_{n \in \mathbb{Z}} sgn(n + \tfrac{R}{2})(n + \tfrac{R}{2}) = -\frac{1}{6} - \frac{1}{4}R(R - 2),$$

$$\sum_{n \in \mathbb{Z}} sgn(n + \tfrac{R}{2})(n + \tfrac{R}{2})^2 = -\frac{1}{12}R(R - 1)(R - 2).$$

## 3.2 The contribution to the partition function

We have established that the theory on $S^1$ is gapped in infinite volume. (The gap of the zero modes develops non-perturbatively.) We now compute the contribution from the zero modes to the generating function. After we have understood the zero modes, we will proceed to make a prediction for the $\beta^{-2}, \beta^{-1}, \beta^0$ terms in the small $\beta$ limit. (As we explained above, due to the zero modes being trivially gapped, it is not actually necessary to analyze them in order to obtain predictions for the $\beta^{-2}, \beta^{-1}$ terms. But we prefer to proceed in this way.)

We first record the result for the case of level $k$ $U(N)$ vector multiplet on the squashed three-sphere $S^3_b$, where $b$ denotes the squashing parameter (see Section 6 for explicit formulae describing this geometry). The gauge kinetic term is $\mathcal{Q}$-exact and therefore the answer is independent of the gauge coupling. The result for the $S^3_b$ partition function, is given by (see the review [72] and references therein)

$$Z_{U(N)_k} = \frac{1}{N!} \int_{-\infty}^{\infty} \prod_{i=1}^{N} d\lambda_i \, e^{i\pi k \sum_{i=1}^{N} \lambda_i^2} \prod_{j>i} 4 \sinh(\pi b \lambda_{ij}) \sinh(\pi b^{-1} \lambda_{ij}), \tag{3.3}$$

where $\lambda_{ij} = \lambda_i - \lambda_j$. This integral can be performed (see [84, 85]) with the aid of the Weyl denominator formula:

$$\prod_{j>i} 2 \sinh(\pi b^{\pm 1} \lambda_{ij}) = \sum_{\sigma} (-1)^{\sigma} \prod_{j} e^{2\pi b^{\pm 1}(\frac{N+1}{2} - \sigma(j))\lambda_j} \, .$$

The integral can be now done explicitly and we find

$$Z_{U(N)_k} = \frac{1}{N!} (-ik)^{-N/2} e^{-\frac{\pi i}{12k} N(N+1)[6(N+1)-(b^2+b^{-2})(N-1)]} \sum_{\sigma_1, \sigma_2} (-1)^{\sigma_1 + \sigma_2} e^{\frac{2\pi i}{k} \sum_j \sigma_1(j)\sigma_2(j)} .$$

Simplifying the last remaining sum with the Weyl denominator formula again we find

$$\sum_{\sigma_1, \sigma_2} (-1)^{\sigma_1 + \sigma_2} e^{\frac{2\pi i}{k} \sum_j \sigma_1(j)\sigma_2(j)} = N! \, e^{\frac{2\pi i}{k} \frac{N(N+1)^2}{4}} (i)^{N(N-1)/2} \prod_{j>l} 2 \sin\left((j-l)\frac{\pi}{k}\right).$$

Combining these terms together we finally find:

$$Z_{U(N)_k} = k^{-N/2} e^{\frac{\pi i}{12k} N(N^2-1)(b^2+b^{-2})} i^{N^2/2} \prod_{j>l} 2 \sin\left((j-l)\frac{\pi}{k}\right) . \tag{3.4}$$

We are ultimately interested in the $SU(N)_k$ partition function. It is given by

$$Z_{SU(N)_k} = \frac{1}{N!} \int_{-\infty}^{\infty} \prod_{i=1}^{N-1} d\lambda_i \, e^{i\pi k \sum_{i=1}^{N} \lambda_i^2} \prod_{j>i} 4 \sinh(\pi b \lambda_{ij}) \sinh(\pi b^{-1} \lambda_{ij}) , \tag{3.5}$$

where $\lambda_N = -\sum_{i=1}^{N-1} \lambda_i$. It is easy to evaluate this integral once the $U(N)_k$ case is known by a simple change of variables, shifting all the eigenvalues other than $\lambda_N$ by $\sum_{i=1}^{N} \lambda_i$. We find $Z_{SU(N)_k} = \sqrt{\frac{k}{N}} Z_{U(N)_k}$. As a result, we can summarize that

$$Z_{SU(N)_k} = \sqrt{\frac{1}{N}} k^{-(N-1)/2} e^{\frac{\pi i}{12k} N(N^2-1)(b^2+b^{-2})} i^{\frac{N^2-1}{2}} \prod_{j>l} 2 \sin\left((j-l)\frac{\pi}{k}\right) . \tag{3.6}$$

Let us see what are some of the consequences of (3.6). First, we see that $k=0$ seems to make little sense. This is in line with what we reviewed above: this theory has no vacuum. If

$0 < |k| \le N-1$ then $\prod_{j>l} 2\sin\left((j-l)\frac{\pi}{k}\right) = 0$. The vanishing of the partition function should be interpreted as a signal of spontaneous supersymmetry breaking, again in agreement with what we explained before. Technically, the partition function vanishes because the theory has a massless goldstino in flat space, but due to it being a Nambu-Goldstone fermion it is not conformally coupled to the sphere and hence there is a zero mode of the corresponding Dirac operator on the sphere (the fact that the goldstino must be accompanied by a nontrivial TFT is not visible in this computation due to the zero mode). For $|k| > N$ the theory flows to a supersymmetric vacuum with a a topological field theory. This explains why the real part of $\log Z_{SU(N)_k}$ is nonzero in this case (in fact, for $k > N$, it is just the $f$-coefficient of the $SU(N)_{k-N}$ topological field theory, to which the supersymmetric theory flows). $k = \pm N$ is a special case and it happens to be the case most interesting to us due to (3.2). For instance, setting $k = N$ and using $\prod_{j>l} 2\sin\left((j-l)\frac{\pi}{N}\right) = N^{N/2}$ we find

$$Z_{SU(N)_N} = e^{\frac{\pi i}{12}(N^2-1)(b^2+b^{-2})+\frac{\pi i}{4}(N^2-1)} \,. \tag{3.7}$$

That this is a pure phase is indicative of the fact that this theory flows to a trivially gapped phase (since the corresponding $f$-coefficient vanishes). The gauge fields are confined and decouple in the infrared and the gauginos are likewise massive due to the Chern-Simons term.

The phases in (3.7) can be interpreted as Chern-Simons contact terms for background fields. To elucidate that, let us consider in a little more detail the case of $k = N$ (the case of $k = -N$ is entirely analogous). Since the theory flows to a gapped trivial phase the infrared contact terms can be understood from integrating out the gauginos at one loop. (There are no nontrivial Hall conductivities.) They lead to a Chern-Simons term $I'_3$ for the $R$-symmetry gauge field with coefficient $-(N^2-1)/2$ and they also shift the gravitational Chern-Simons term $I'_4$ as $N^2-1$ Dirac fermions would do. Since the gauginos are un-charged under the KK photon, we can ignore it. The dependence on squashing from the $R$-symmetry contact term is $e^{\frac{\pi i}{8}(N^2-1)(b^2+b^{-2}+2)}$, and from the gravitational Chern-Simons term it is $e^{-\frac{\pi i}{24}(N^2-1)(b^2+b^{-2}+v)}$ (here we used the basis of contact-terms of (2.8)). Combining these together we obtain the phase

$$e^{\frac{\pi i}{12}(N^2-1)(b^2+b^{-2}+3-v/2)} \,,$$

which nicely agrees with the result found by localization (3.7) upon setting $v = 0$. As already mentioned, the choice $v = 0$ arises naturally for a frame adapted to the supersymmetry of the background.

A simple way to summarize these results is to say that the zero modes of the gauginos behave as if they have a *negative* real mass and the zero modes of the vector fields confine and decouple (they do not contribute to any Chern-Simons contact terms for background fields).

# 4 Combining with massive mode contributions

Armed with our understanding of the zero modes, with the simple conclusion being that we need to treat the fermion zero modes as if they have a negative mass and the gauge fields can be ignored altogether due to confinement, we can now compute all the supersymmetrized background Chern-Simons terms in the effective theory from integrating out the massive KK and (tree-level) zero modes. We have in total four Chern-Simons terms in the effective theory and we need to evaluate each of them as follows:

- The coefficient of the KK photon Chern-Simons term (2.4) from a chiral multiplet with $R$-charge $R_I$ is

$$\frac{1}{2}\sum_n (n+R_I/2)^2 sgn(n+R_I/2) = -\frac{1}{24}R_I(R_I-1)(R_I-2) \,.$$

We need to sum over all the chiral multiplets in the theory with their respective $R$-charges. The vector multiplet makes a similar contribution provided we substitute $R_I = 2$, so that $R_I - 1 = 1$ as required for the gaugino. Since the coefficient above vanishes for the zero mode, we need not worry about it. In fact the whole contribution of the vector multiplet vanishes. (This can be explained in simple terms – the spectrum of the gauginos is vector-like and hence parity preserving.) Therefore, the total KK photon Chern-Simons term coefficient is $-\frac{1}{24} \sum_I R_I (R_I - 1)(R_I - 2)$, where we sum over all chiral multiplets. This is in fact identical to $\frac{-1}{24}(\text{Tr}R^3 - \text{Tr}R)$ where the traces are taken over the fermions with their corresponding $R$-charges, which are $R_I - 1$ in the chiral multiplets. Multiplying it by $I_1$ given in (2.9), this contributes to the partition function as

$$\log Z = -\frac{\pi^3 i \ell^2}{6\beta^2}(\text{Tr}R^3 - \text{Tr}R) . \tag{4.1}$$

This is squashing independent.

- The coefficient of the mixed KK photon-$R$-symmetry Chern-Simons term (2.5) from a chiral multiplet with $R$ charge $R_I$ is

$$\frac{R_I - 1}{2} \sum_n (n + R_I/2) sgn(n + R_I/2) = -\frac{R_I - 1}{12} - \frac{1}{8} R_I(R_I - 1)(R_I - 2),$$

and we need to sum over all chiral multiplets along with the vector multiplet, which again gives a contribution identical to the one above with $R_I = 2$. Note an important thing: the gaugino zero mode does not contribute since it has zero KK charge (however, the gaugino nontrivial KK modes do contribute). The above combination is observed to be identical to $-\frac{1}{24}(3\text{Tr}R^3 - \text{Tr}R)$. Hence, using $I_2$ in (2.9), the partition function receives a contribution from this as

$$\log Z = -\frac{\pi^2 \ell}{12\beta}(b + b^{-1})(3\text{Tr}R^3 - \text{Tr}R). \tag{4.2}$$

- The coefficient of the $U(1)_R$-$U(1)_R$ Chern-Simons term (2.6) (in the basis (2.8)) receives a contribution from a chiral multiplet with $R$-charge $R_I$ as

$$\frac{(R_I - 1)^2}{2} \sum_n sgn(n + R_I/2) = -\frac{(R_I - 1)^3}{2} .$$

The gauginos have to be accounted for very carefully due to the gaugino zero mode. Let us first compute the contribution from the non-zero modes. They are clearly vector-like (symmetric about zero mass) and all the nontrivial KK modes have the same $R$-charge and hence they give a vanishing contribution.[13] Lastly we have to consider the contribution of the gaugino zero mode. We have shown that quantum effects lift the gaugino zero mode and effectively make it behave as if it was a massive particle with a negative mass. Therefore it contributes to the $U(1)_R$-$U(1)_R$ Chern-Simons term as $-(N^2 - 1)/2$. Combining these results we see that it matches $-\frac{1}{2}\text{Tr}R^3$ and multiplying it by $I_3$ in (2.9) results in a contribution to the partition function as

$$\log Z = \frac{\pi i}{8}(b + b^{-1})^2 \text{Tr}R^3 . \tag{4.3}$$

---

[13]This can be justified mathematically from $\sum_{n \in \mathbb{Z}} sgn(n + R_I/2) = 1 - R_I$. Of course, this formula only makes sense in some fundamental domain, say $R_I \in [0, 2)$ and the function is periodic otherwise. In particular, at $R_I = 2$ there is a discontinuity where from the left the function approaches $-1$ and from the right it approaches $+1$. This jump is in accord with one eigenvalue crossing zero from below. Removing the contribution of that particular eigenvalue either for $R_I = 2^-$ or for $R_I = 2^+$ we find that the rest contribute 0. Therefore it is meaningful to say that $\sum_{n \neq 0} sgn(n) = 0$.

- Finally we need to consider the gravitational Chern-Simons term. From the chiral multiplets we have

$$\sum sgn(n + R_I/2) = -(R_I - 1) \,.$$

Note that there is no factor of $1/2$ since we are integrating out Dirac fermions. The nontrivial KK modes of the gaugino again do not make a contribution since they are vector-like. The zero modes should be treated due to non-perturbative effects as fermions with negative mass. This gives another contribution which is $-(N^2 - 1)$. The gravitational Chern-Simons term is therefore with coefficient $-\sum_I (R_I - 1) - (N^2 - 1)$, where the sum over $I$ is over all chiral multiplets. This can be summarized as $-\text{Tr}R$. Therefore the contribution to the partition function is

$$\log Z = -\frac{\pi i}{24}(b^2 + b^{-2})\,\text{Tr}R \,,$$

where we have used the expression for $I_4' = I_4 - \frac{1}{12}I_3$, cf. (2.9), and fixed the framing dependence as $\nu = 0$.

Adding up all contributions, including the $\log|G|$ contribution from the degeneracy of vacua, we obtain:

$$\log \mathcal{I} = -\frac{\pi^3 i \ell^2}{6\beta^2}(\text{Tr}R^3 - \text{Tr}R) - \frac{\pi^2 \ell}{12\beta}(b + b^{-1})(3\text{Tr}R^3 - \text{Tr}R) + \frac{\pi i}{8}(b + b^{-1})^2 \,\text{Tr}R^3$$

$$- \frac{\pi i}{24}(b^2 + b^{-2})\,\text{Tr}R + \log|G| + O(\beta)\,. \tag{4.4}$$

We now relate this to the behavior of the superconformal index in four dimensions, which is a holomorphic function of $\omega_{1,2}$. To deduce this holomorphic function from the above discussion we recall that $\text{Re}\,\omega_1 = \beta b/\ell$ and $\text{Re}\,\omega_2 = \beta/(b\ell)$. We can rewrite (4.4) in terms of $\text{Re}\,\omega_1$, $\text{Re}\,\omega_2$ and then remove the $\text{Re}$ sign to obtain analytic functions in two variables. Doing so, we obtain precisely the expression given in (1.7).

Since the EFT is trivially gapped all further terms contributing to the small-$\beta$ expansion of $\log Z$ should be contact terms. If our conjecture that these are all true $D$ terms is valid, we can extend our result to all polynomial order in $\beta$ as discussed in the Introduction, and reach the result (1.8).

## 5 Examples

### 5.1 Free chiral multiplet

Let us consider a single free chiral multiplet with $R$-charge $0 < r < 2$. The supersymmetric index is given in terms of the elliptic Gamma function as

$$\mathcal{I}_{\text{chiral}} = \Gamma\left(e^{-\frac{r}{2}(\omega_1 + \omega_2 + 2\pi i n_0)}, e^{-\omega_1}, e^{-\omega_2}\right), \tag{5.1}$$

with $\text{Re}\,\omega_1 > 0$, $\text{Re}\,\omega_2 > 0$. The integer $n_0$ distinguishes the different sheets: $n_0 = 0$ is the first sheet, while $n_0 = 1$ leads us to the second sheet and $n_0 = -1$ to its "complex conjugate" sheet.

Let us fix

$$\frac{\omega_1}{\omega_2} \in \mathbb{R} \qquad \text{and} \qquad n_0 = \pm 1\,. \tag{5.2}$$

Then we can apply an asymptotic formula for the elliptic Gamma function (see [86, Prop. 2.11] or [23]), implying that in the limit $\omega_{1,2} \to 0$,

$$\log \mathcal{I}_{\text{chiral}} = \frac{(\omega_1 + \omega_2 + 2\pi i n_0)^3}{48\omega_1\omega_2}(r-1)^3 - \frac{(\omega_1 + \omega_2 + 2\pi i n_0)(\omega_1^2 + \omega_2^2 - 4\pi^2)}{48\omega_1\omega_2}(r-1)$$
$$+ O(e^{-\ell/\beta}). \tag{5.3}$$

This expression agrees with (1.8), with $|G| = 1$ since this theory does not have a one-form symmetry.

For $\omega_1/\omega_2 \notin \mathbb{R}$, we can still apply a slightly less accurate estimate [86, Prop. 2.12], which only ensures control on the diverging terms in the limit.

## 5.2 A theory with just one sheet

We now discuss a peculiar example where the first and second sheet are the same.[14]

Consider an $\mathcal{N} = 1$ theory with gauge group $SU(3) \times SU(3)$ and with 9 chiral multiplets in the $(\mathbf{3}, \mathbf{0})$ representation, 9 in the $(\mathbf{0}, \bar{\mathbf{3}})$ and 3 in the bifundamental representation $(\bar{\mathbf{3}}, \mathbf{3})$. The superconformal $R$-charge is $R = \frac{2}{3}$. One can check that all fermionic gauge-invariant operators have $R$-charge $R = 1$ (mod 2), while all bosonic gauge-invariant operators have $R$-charge $R = 2$ (mod 2), hence $e^{-\pi i R} = (-1)^F$. This implies that the first and the second sheet of the index are the same. In particular, the Cardy limits on the first and second sheet must give the same result. In fact, the two sheets must be related by a gauge transformation, i.e. a transformation shifting the holonomies.

This poses some puzzles. This theory has $\text{Tr}R < 0$ and hence one might expect (1.3) to hold. But on the other hand, we claimed that the result on the second sheet (1.7) holds very generally. These results clearly disagree. The resolution is very simple: Since the theory is not vector like (i.e. it is chiral) in fact $\text{Tr}R < 0$ is not sufficient to guarantee that (1.3) holds. We will show below explicitly that the holonomy vacuum on the first sheet is away from the origin. When considering the properties of that vacuum, we find exact agreement with the most singular piece in the prediction (1.7). (We did not try to go beyond the most singular piece.)

The Cardy limit of the index is controlled by an effective potential for the gauge holonomies $e^{2\pi i u_i}$, $i = 1, \ldots, \text{rank} \, G_{\text{gauge}}$,

$$\mathcal{I} \propto \int du \, e^{-V^{\text{eff}}(u)}. \tag{5.4}$$

This takes an a priori different form on the first sheet, on the second sheet and on the complex conjugate sheet. One has [34]

$$V^{\text{eff}} = -\frac{2\pi^3 i}{3\omega_1\omega_2} \sum_{I \in \text{chirals}} \sum_{\rho_I \in \mathcal{R}_I} \kappa\left(\rho_I \cdot u - n_0 \frac{r_I}{2}\right) + O(\beta^{-1}), \tag{5.5}$$

where $I$ labels the chiral fields in the theory, and $\rho_I$ are the weights of the representation $\mathcal{R}_I$ in which the $I$-th field transforms. The function $\kappa$ is given by

$$\kappa(x) = \{x\}(1 - \{x\})(1 - 2\{x\}), \tag{5.6}$$

with $\{x\} = x - \lfloor x \rfloor$ being the fractional part. Note that $\kappa(-x) = -\kappa(x)$. Again the integer $n_0$ distinguishes the different sheets.

---

[14]We thank S. Razamat for many discussions about such theories.

It was proven in [34] that under mild assumptions on the $R$-charges, for $n_0 = \pm 1$ there is a saddle at $u_i = 0$, $i = 1, \ldots, \text{rank } G_{\text{gauge}}$, which leads to the estimate for the index (assuming there are no other saddles that dominate over the one at the origin) in agreement with (1.7)

$$\log \mathcal{I} = \mp \frac{\pi^3 i}{6\omega_1 \omega_2} (\text{Tr} R^3 - \text{Tr} R) + O(\beta^{-1}), \qquad \text{for } n_0 = \pm 1. \tag{5.7}$$

The estimate with $n_0 = 1$ is valid in the regime of chemical potentials $\text{Re}\left(\frac{i}{\omega_1 \omega_2}\right) < 0$, while the estimate with $n_0 = -1$ is valid in the opposite regime, $\text{Re}\left(\frac{i}{\omega_1 \omega_2}\right) > 0$. Note that in our description this requires at least one of the twisting parameters $k_1, k_2$ to be non-vanishing, so that either $\omega_1$ or $\omega_2$ has a non-vanishing imaginary part.

For the theory at hand, $\text{Tr} R^3 = 3$ and $\text{Tr} R = -21$, so we obtain

$$\log \mathcal{I} = \mp \frac{4\pi^3 i}{\omega_1 \omega_2} + O(\beta^{-1}), \qquad \text{for } n_0 = \pm 1. \tag{5.8}$$

The general analysis of the effective potential on the first sheet, obtained by setting $n_0 = 0$ in (5.5), can be found in [22, 24].[15] For theories with charge-conjugation symmetry, namely for theories such that for any weight $\rho$ there is an opposite weight $-\rho$, the $O(\beta^{-2})$ term in (5.5) with $n_0 = 0$ vanishes identically and the potential is controlled by the subleading $O(\beta^{-1})$ term. This is the situation on which the authors of [22, 24] mostly focused their attention. However, the example we are considering here has no such charge-conjugation symmetry, and the effective potential has a non-vanishing $O(\beta^{-2})$ term even on the first sheet.

We denote by $u_1, u_2$ the variables parameterizing the gauge holonomies of the first $SU(3)$ and by $u_1', u_2'$ those of the second $SU(3)$. These are all taken in the fundamental domain $[0, 1)$. We also introduce $u_3 = -u_1 - u_2$ and $u_3' = -u_1' - u_2'$ for convenience. Then the $n_0 = 0$ potential involves the function

$$\sum_{I \in \text{chirals}} \sum_{\rho_I \in \mathcal{R}_I} \kappa(\rho_I \cdot u) = 9 \sum_{i=1}^3 \kappa(u_i) + 9 \sum_{i=1}^3 \kappa(-u_i') + 3 \sum_{i,j=1}^3 \kappa(-u_i + u_j'). \tag{5.9}$$

We observe that shifting

$$u_i \to u_i + \frac{2}{3}, \qquad u_i' \to u_i' + \frac{1}{3}, \tag{5.10}$$

yields exactly the $n_0 = 1$ potential, while the shift

$$u_i \to u_i + \frac{1}{3}, \qquad u_i' \to u_i' + \frac{2}{3} \tag{5.11}$$

gives the potential on the $n_0 = -1$ sheet. It follows that the saddles at the origin that are found on the $n_0 = \pm 1$ sheets are mapped into saddles at non-trivial values of the gauge holonomies in the $n_0 = 0$ sheet. The two descriptions are equivalent and the physics is the same, just occuring at different VEVs of the holonomies.

It should not be hard to carry out the analysis beyond the most singular term in the $1/\beta$ expansion.

It was shown in [34] that the saddle at the origin is the dominant one for quiver gauge theories with charge-conjugation symmetry and all $R$-charges being between 0 and 1. A numerical study of the effective potential shows that the present theory is an example of a chiral theory where the saddle at the origin also dominates the Cardy limit for $n_0 = \pm 1$.

The example discussed here is obtained by taking the E-string theory on a genus 2 surface, leading to the above 4d theory. This construction admits a generalization to the E-string on a genus $g$ surface. All of these theories will have the same property of having one sheet [87–89] (see Figure 1 in the latter reference for the quiver of genus $g$).

---

[15]See Eq. (2.28) in [24]. There the chemical potentials $\omega_1, \omega_2$ are taken real, here we are very slightly extending that analysis to a twisted $S^1 \times S_b^3$ background, which as we show in Section 6 gives complex chemical potentials.

# 6 General twisted background

In this section, we define a supersymmetric 4d background of $S^1 \times S^3$ topology that encodes generically complex parameters $\omega_1, \omega_2$. We then reduce it along $S^1$, obtain a supersymmetric background of 3d new-minimal supegravity, and evaluate the relevant contact terms.

## 6.1 The background

In four dimensions, supersymmetric backgrounds with two supercharges of opposite R-charge are constructed by solving the "new-minimal equations",[16]

$$\left(\nabla_\mu - iA_\mu + iV_\mu + iV^\nu \sigma_{\mu\nu}\right)\zeta = 0\,,$$
$$\left(\nabla_\mu + iA_\mu - iV_\mu - iV^\nu \widetilde{\sigma}_{\mu\nu}\right)\widetilde{\zeta} = 0\,, \tag{6.1}$$

where $\zeta$ and $\widetilde{\zeta}$ are two-component spinors of opposite chirality and opposite R-charge, which represent the parameters of the supersymmetry transformations. In addition to the metric, one has a background gauge field $A_\mu$, coupling to the $R$-current, and a globally well-defined background one-form $V_\mu$.

We consider a space with $S^1 \times S^3$ topology, parameterized by coordinates $\tau \sim \tau + \beta$ on $S^1$ and $(\theta, \varphi_1, \varphi_2)$ on $S^3$, with $\varphi_1 \sim \varphi_1 + 2\pi$, $\varphi_2 \sim \varphi_2 + 2\pi$ and $\theta \in (0, \frac{\pi}{2})$. We take the metric

$$\begin{aligned}
\mathrm{d}s^2 = \ &\Omega(\theta)^2 \Big[ \mathrm{d}\tau^2 + \ell^2 \left( b^2 \cos^2\theta + b^{-2}\sin^2\theta \right)\mathrm{d}\theta^2 \\
&+ b^{-2}\cos^2\theta \left(\ell\,\mathrm{d}\varphi_1 + k_1\mathrm{d}\tau\right)^2 + b^2\sin^2\theta \left(\ell\,\mathrm{d}\varphi_2 + k_2\mathrm{d}\tau\right)^2 \Big]\,,
\end{aligned} \tag{6.2}$$

where the conformal factor $\Omega$ is any smooth positive function of $\theta$; later we will specify a convenient choice. The real parameter $b > 0$ controls the squashing of $S^3$, while the real parameters $k_1, k_2$ specify the twisting of $S^3$ over $S^1$ and $\ell$ is the length scale of $S^3$.[17] When $b = 1$, $k_1 = k_2 = 0$, the metric describes a space conformal to the direct product of $S^1$ with a round $S^3$. For $k_1 = k_2 = 0$ but $b \neq 1$, the space is conformal to the direct product of $S^1$ with an elliptically squashed three-sphere, denoted by $S_b^3$; this is the background considered in the previous sections.

The metric admits the complex Killing vector

$$K = \frac{1}{2}\left[-i\frac{\partial}{\partial\tau} + \ell^{-1}(b + ik_1)\frac{\partial}{\partial\varphi_1} + \ell^{-1}(b^{-1} + ik_2)\frac{\partial}{\partial\varphi_2}\right]\,. \tag{6.3}$$

As a one-form, $K$ reads

$$K = \frac{1}{2}\Omega^2\left[b^{-1}\cos^2\theta \left(\ell\,\mathrm{d}\varphi_1 + k_1\mathrm{d}\tau\right) + b\,\sin^2\theta \left(\ell\,\mathrm{d}\varphi_2 + k_2\mathrm{d}\tau\right) - i\,\mathrm{d}\tau\right]\,. \tag{6.4}$$

This satisfies

$$K_\mu K^\mu = 0\,, \qquad \text{and} \qquad K^\nu \nabla_\nu \overline{K}^\mu - \overline{K}^\nu \nabla_\nu K^\mu = 0\,. \tag{6.5}$$

From the general discussion in [11, 90],[18] these properties are sufficient to ensure that any $\mathcal{N} = 1$ field theory with an $R$-symmetry can be defined in the curved space under consideration

---

[16]We use the conventions of [9].

[17]Note that $k_1, k_2$ may be removed by shifting the angular coordinates as $\varphi_1 = \check{\varphi}_1 - k_1\tau/\ell$, $\varphi_2 = \check{\varphi}_2 - k_2\tau/\ell$. However in this case the periodic identifications of the coordinates would be twisted, that is when making a revolution around $S^1$ we would have the identification $(\tau \sim \tau + \beta$, $\check{\varphi}_1 \sim \check{\varphi}_1 + \beta k_1/\ell$, $\check{\varphi}_2 \sim \check{\varphi}_2 + \beta k_2/\ell)$. We prefer to work with standard identifications of the coordinates.

[18]In particular, see Sect. 4.2 of [11]. A background similar to this one is discussed in [9, App. D].

while preserving two supercharges of opposite $R$-charge, meaning that both equations (6.1) admit a non-vanishing solution. This requires to choose the background fields as

$$
V = \frac{i\,\mathrm{d}\tau}{\ell\sqrt{b^2\cos^2\theta + b^{-2}\sin^2\theta}} + \mathrm{d}x^\mu J_\mu{}^\nu \nabla_\nu \log\Omega + \kappa(\theta)K\,,
$$

$$
A = \frac{1}{2\ell\sqrt{b^2\cos^2\theta + b^{-2}\sin^2\theta}}\Big[2i\,\mathrm{d}\tau - b^{-1}(\ell\,\mathrm{d}\varphi_1 + k_1\mathrm{d}\tau) - b\,(\ell\,\mathrm{d}\varphi_2 + k_2\mathrm{d}\tau)\Big]
$$

$$
+ \frac{1}{2}\,(\mathrm{d}\varphi_1 + \mathrm{d}\varphi_2) + \frac{3}{2}\,\mathrm{d}x^\mu J_\mu{}^\nu \nabla_\nu \log\Omega + \frac{3}{2}\kappa(\theta)K\,. \tag{6.6}
$$

The function $\kappa(\theta)$ is arbitrary and will be fixed later. We have fixed the gauge of $A$ so as to ensure regularity at the poles of $S^3$.[19]

We pick the frame

$$
e^1 = \ell\,\Omega\,\sqrt{b^2\cos^2\theta + b^{-2}\sin^2\theta}\,\mathrm{d}\theta\,,
$$

$$
e^2 = \Omega\,\sin\theta\cos\theta\,\big(b^{-1}(\ell\,\mathrm{d}\varphi_1 + k_1\mathrm{d}\tau) - b\,(\ell\,\mathrm{d}\varphi_2 + k_2\mathrm{d}\tau)\big)\,,
$$

$$
e^3 = \Omega\big(b^{-1}\cos^2\theta\,(\ell\,\mathrm{d}\varphi_1 + k_1\mathrm{d}\tau) + b\,\sin^2\theta\,(\ell\,\mathrm{d}\varphi_2 + k_2\mathrm{d}\tau)\big)\,,
$$

$$
e^4 = \Omega\,\mathrm{d}\tau \tag{6.7}
$$

and define the volume form as $\mathrm{vol}_4 = e^1\wedge e^2\wedge e^3\wedge e^4$. The frame is chosen so that $K = \frac{1}{2}\Omega\,(e^3 - i\,e^4)$. Then one can introduce the self-dual and anti-self-dual two-forms

$$
J = -e^1\wedge e^2 - e^3\wedge e^4\,, \qquad \widetilde{J} = e^1\wedge e^2 - e^3\wedge e^4\,, \tag{6.8}
$$

and show that $J^\mu{}_\nu$ and $\widetilde{J}^\mu{}_\nu$ are commuting integrable complex structures. The vector $K^\mu$ is holomorphic with respect to both of them,

$$
J^\mu{}_\nu K^\nu = iK^\mu\,, \qquad \widetilde{J}^\mu{}_\nu K^\nu = iK^\mu\,. \tag{6.9}
$$

In the chosen frame, the spinorial parameters solving the supersymmetry conditions (6.1), read

$$
\zeta = \sqrt{\frac{\Omega}{2}}\,\mathrm{e}^{\frac{i}{2}(\varphi_1+\varphi_2)}\begin{pmatrix}0\\1\end{pmatrix}\,, \qquad \widetilde{\zeta} = \sqrt{\frac{\Omega}{2}}\,\mathrm{e}^{-\frac{i}{2}(\varphi_1+\varphi_2)}\begin{pmatrix}1\\0\end{pmatrix}\,. \tag{6.10}
$$

The supersymmetry transformations obtained from new-minimal supergravity on the background above give the algebra

$$
\{\delta_\zeta, \delta_{\widetilde{\zeta}}\} = 2i\big(\mathcal{L}_K - iR\,K^\mu A_\mu\big)\,, \tag{6.11}
$$

$$
\delta_\zeta^2 = \delta_{\widetilde{\zeta}}^2 = 0\,, \tag{6.12}
$$

where $\mathcal{L}_K$ is the Lie derivative along $K$, and $R$ is the $R$-charge of the field on which the algebra is represented.

**Complex structure moduli.** We have seen that our background space is complex. Every complex manifold with $S^1 \times S^3$ topology is a primary Hopf surface, and our background qualifies as a primary Hopf surface of the first type, see e.g. [8,9] and references therein. These are quotients of $\mathbb{C}^2 - (0,0)$ where the coordinates $(z_1, z_2)$ are identified as

$$
(z_1, z_2) \sim (pz_1, qz_2)\,, \tag{6.13}
$$

---

[19]For $\theta \to 0$ the differential $\mathrm{d}\varphi_2$ is not well-defined, so one needs to make sure that $A_{\varphi_2} \to 0$; analogously for $\theta \to \frac{\pi}{2}$, $\mathrm{d}\varphi_1$ is not well-defined and one needs $A_{\varphi_1} \to 0$.

with $p, q$ being complex parameters satisfying $0 < |p| \leq |q| < 1$. These are precisely the complex structure moduli of the Hopf surface.

We now show that for our background the complex structure moduli are given by

$$p = e^{-\omega_1}, \qquad q = e^{-\omega_2} \tag{6.14}$$

with

$$\omega_1 = \frac{\beta}{\ell}(b + ik_1), \qquad \omega_2 = \frac{\beta}{\ell}(b^{-1} + ik_2). \tag{6.15}$$

We introduce complex coordinates in our $S^1 \times S^3$ space so that (6.13) is manifest. We take

$$z_1 = \frac{\cos\theta}{1 + b\sqrt{b^2\cos^2\theta + b^{-2}\sin^2\theta}} \, e^{b\sqrt{b^2\cos^2\theta + b^{-2}\sin^2\theta}} \, e^{-i\varphi_1 - (b + ik_1)\tau/\ell},$$

$$z_2 = \frac{\sin\theta}{1 + b^{-1}\sqrt{b^2\cos^2\theta + b^{-2}\sin^2\theta}} \, e^{b^{-1}\sqrt{b^2\cos^2\theta + b^{-2}\sin^2\theta}} \, e^{-i\varphi_2 - (b^{-1} + ik_2)\tau/\ell}. \tag{6.16}$$

These coordinates are chosen so that $\widetilde{J}_\mu{}^\nu \partial_\nu z_1 = i\partial_\mu z_1$, $\widetilde{J}_\mu{}^\nu \partial_\nu z_2 = i\partial_\mu z_2$, namely they are holomorphic with respect to the complex structure $\widetilde{J}$.[20] One can see that when $\tau$ is not compactified they parameterize $\mathbb{C}^2 - (0, 0)$. Indeed, for fixed $\theta, \tau$ we see that $\varphi_1, \varphi_2$ are polar angles for the two complex planes in $\mathbb{C}^2$; moreover, for fixed $|z_2|$, one has that $|z_1|$ covers the positive real numbers, and vice-versa. The important point for us is that making the identification $\tau \sim \tau + \beta$ corresponds to identifying

$$(z_1, z_2) \sim (e^{-\frac{\beta}{\ell}(b + ik_1)}z_1, \, e^{-\frac{\beta}{\ell}(b^{-1} + ik_2)}z_2). \tag{6.17}$$

Comparing with (6.13), this shows that

$$p = e^{-\frac{\beta}{\ell}(b + ik_1)} = e^{-\omega_1}, \qquad q = e^{-\frac{\beta}{\ell}(b^{-1} + ik_2)} = e^{-\omega_2} \tag{6.18}$$

are the complex structure parameters of our Hopf surface. Notice that the condition $0 < |p| \leq |q| < 1$ is satisfied by taking $0 < \operatorname{Re}\omega_2 \leq \operatorname{Re}\omega_1$, that is $b \geq 1$. $\omega_1, \omega_2$ are otherwise arbitrary complex parameters. Although the background under study is not unique (see e.g. [9] for more general choices including arbitrary functions), the one considered here encodes the most general complex structure and is still simple enough to allow for a completely explicit treatment.

Before continuing with the reduction to 3d it may be useful to pause and make a few comments. We emphasize that even if the complex-structure parameters $\omega_1, \omega_2$ take complex values, our background metric (6.2) is *real*. Some other descriptions leading to complex $\omega_1, \omega_2$ have considered a background metric with *complex* components, arising as the boundary metric of a complexified section of a black hole solution to five-dimensional supergravity [27, 91]. It would be interesting to understand if there is a geometric relation between these two descriptions.

Notice that the Killing spinors (6.10) are independent of the $S^1$ coordinate. These satisfy the usual supersymmetric boundary conditions imposing that all (dynamical and background) fields are periodic around $S^1$, as well as our twisted boundary conditions (1.6). So we have a good supersymmetric background in both cases. Of course, the path integral depends on the background as well as on the boundary conditions, hence it is not the same in the two cases.

If we start with the background above and periodic boundary conditions around $S^1$, the twisted boundary conditions (1.6) corresponding to the index on the second sheet can be obtained as follows. From (6.2), (6.6) we see that the imaginary shift $\omega_1 \to \omega_1 + 2\pi i$ (that is

---

[20]We thank P. Bomans for pointing out a choice of coordinates adapted to our complex structure.

$k_1 \rightarrow k_1 + \frac{2\pi\ell}{\beta}$) can be reabsorbed by the change of coordinate $\varphi_1 \rightarrow \varphi_1 - \frac{2\pi}{\beta}\tau$, accompanied by an $R$-symmetry transformation $A \rightarrow A + \frac{\pi}{\beta}d\tau$. The combination of these transformations leaves the background invariant but alters the boundary conditions of all fields around $S^1$ as in (1.6).

Alternatively, we could have implemented the shift $\omega_1 \rightarrow \omega_1 + 2\pi i$ leading to the second sheet by maintaining periodic boundary conditions for all fields and allowing the background fields (6.2), (6.6) to simply transform according to $k_1 \rightarrow k_1 + \frac{2\pi\ell}{\beta}$. Yet another description is obtained by partially untwisting the boundary conditions (1.6) via the transformation $A \rightarrow A - \frac{\pi}{\beta}d\tau$, which would leave us with periodic bosons and anti-periodic fermions. The latter configuration is closely related to the one derived in [27] by studying the asymptotics of the supersymmetric black hole in AdS$_5$. However in this picture a supersymmetry-preserving KK reduction to 3d is less straightforward, as the supercharges depend on the $S^1$ coordinate, so we do not discuss it any further.

Of course, these alternative descriptions are based on the equivalence (up to anomalies) in representing a chemical potential as twisted boundary conditions or as the holonomy for a background gauge field. Indeed, any chemical potential $\mu$ for a charge $Q$, appearing in the partition function as $Z = \mathrm{Tr}\, e^{-\beta(H-\mu Q)}$, corresponds to the twisted identifications $\chi(\tau + \beta) = (-1)^F e^{-\beta\mu q}\chi(\tau)$. These twisted identifications can be undone by a large gauge transformation $\chi \rightarrow e^{i\lambda q}\chi$ with parameter $\lambda = -i\mu\tau$. After the transformation, the fields obey standard identifications $\chi(\tau + \beta) = (-1)^F \chi(\tau)$, however the background field $A$ gauging the symmetry generated by $Q$ has shifted as $A \rightarrow A - i\mu\, d\tau$, and has thus a different holonomy around $S^1$.

## 6.2 Kaluza-Klein reduction to 3d supergravity

We now reduce the background above along the $S^1$, and match it to 3d supergravity. We consider the Kaluza-Klein ansatz for the metric and the other background fields,

$$
\begin{aligned}
ds^2 &= ds_3^2 + e^{2\Phi}(d\tau + c)^2, \\
A &= \mathcal{A} + A_\tau(d\tau + c), \\
V &= \mathcal{V} + V_\tau(d\tau + c).
\end{aligned}
\tag{6.19}
$$

This gives the 3d metric $ds_3^2$, the KK photon gauge field $c$, the 3d gauge field $\mathcal{A}$, the well-defined 3d one-form $\mathcal{V}$, and the scalar fields $A_\tau, V_\tau, \Phi$, all independent of the $\tau$ coordinate.

On general grounds, the dimensional reduction of the 4d new-minimal gravity multiplet should give the 3d new-minimal gravity multiplet together with the KK photon multiplet, whose bosonic components have been introduced in (2.2), (2.3). In Appendix A we work out the general identification of these 3d supergravity fields with the KK fields (6.19) without assuming that the supersymmetry conditions (6.1) are satisfied. Here instead we exploit the fact that the background of interest does solve the equations (6.1) to simplify the analysis slightly and make contact with the dimensional reduction discussed in the Appendix D of [70]. The 4d background considered in that reference is such that

$$
e^\Phi = 1 \qquad \text{and} \qquad A_\tau = V_\tau.
\tag{6.20}
$$

We can arrange for these conditions by making a suitable choice of the arbitrary functions $\Omega(\theta)$ and $\kappa(\theta)$ that appear in our 4d background (6.2), (6.6). In order to ensure $e^\Phi = 1$ we choose

$$
\Omega = \frac{1}{\sqrt{1 + b^{-2}k_1^2 \cos^2\theta + b^2 k_2^2 \sin^2\theta}},
\tag{6.21}
$$

while $A_\tau = V_\tau$ is obtained by setting

$$\kappa = \frac{2\left(b^{-1}k_1 + bk_2\right)\left(i + b^{-1}k_1 \cos^2\theta + bk_2 \sin^2\theta\right)}{\ell\sqrt{b^2 \cos^2\theta + b^{-2}\sin^2\theta}}. \tag{6.22}$$

We will assume these two choices henceforth. These are not expected to affect the final result, which should depend on the complex structure parameters only. In particular, a change in $\Omega$ does not affect the partition function of a superconformal theory as the super-Weyl anomaly vanishes in the background considered [92].

We find that the KK fields coming from the 4d metric read

$$ds_3^2 = \ell^2\Omega^2\left[\left(b^2\cos^2\theta + b^{-2}\sin^2\theta\right)d\theta^2 + b^{-2}\cos^2\theta\,d\varphi_1^2 + b^2\sin^2\theta\,d\varphi_2^2\right]$$
$$- \ell^2\Omega^4\left(b^{-2}k_1\cos^2\theta\,d\varphi_1 + b^2k_2\sin^2\theta\,d\varphi_2\right)^2,$$
$$e^\Phi = 1,$$
$$c = \ell\,\Omega^2\left(b^{-2}k_1\cos^2\theta\,d\varphi_1 + b^2k_2\sin^2\theta\,d\varphi_2\right), \tag{6.23}$$

while those descending from the 4d auxiliary fields are

$$\mathcal{A} = \frac{1}{2\sqrt{b^2\cos^2\theta + b^{-2}\sin^2\theta}}\Big[-2\Omega^2(i + b^{-1}k_1 + bk_2)(b^{-2}k_1\cos^2\theta d\varphi_1 + b^2k_2\sin^2\theta d\varphi_2)$$
$$+ 3\Omega^2(b^{-1}k_1 + bk_2)(i + b^{-1}k_1\cos^2\theta + bk_2\sin^2\theta)(b^{-1}\cos^2\theta\,d\varphi_1 + b\sin^2\theta\,d\varphi_2)$$
$$+ 3\Omega^2(b^{-2}k_1^2 - b^2k_2^2)\sin^2\theta\cos^2\theta(b^{-1}d\varphi_1 - bd\varphi_2) - b^{-1}d\varphi_1 - bd\varphi_2\Big] + \frac{1}{2}(d\varphi_1 + d\varphi_2),$$
$$\mathcal{V} = \frac{i\Omega^2\left(k_2\cos^2\theta\,d\varphi_1 + k_1\sin^2\theta\,d\varphi_2\right)}{\sqrt{b^2\cos^2\theta + b^{-2}\sin^2\theta}},$$
$$A_\tau = V_\tau = \frac{i + b^{-1}k_1 + bk_2}{\ell\sqrt{b^2\cos^2\theta + b^{-2}\sin^2\theta}}. \tag{6.24}$$

The orientation is specified by the volume form

$$\text{vol}_3 = \ell^3\,\Omega^4\sin\theta\cos\theta\sqrt{b^2\cos^2\theta + b^{-2}\sin^2\theta}\,d\theta \wedge d\varphi_1 \wedge d\varphi_2. \tag{6.25}$$

One can check that [70]

$$\mathcal{V} = -\frac{i}{2} * dc, \tag{6.26}$$

where the Hodge star is computed with the 3d metric and volume form above. This relation is a consequence of supersymmetry of the 4d background (see Appendix A for a proof).

Next we use the dictionary developed in [70, App. D] to identify the auxiliary fields $\check{A}, \check{V}, H$ in the 3d new-minimal supergravity multiplet. These are given by

$$\check{V} = 2\mathcal{V},$$
$$\check{A} = \mathcal{A} + \mathcal{V},$$
$$H = A_\tau = V_\tau. \tag{6.27}$$

We also find that the KK photon multiplet is given by

$$\text{KK photon multiplet}: \left(\sigma_{\text{kk}} = -1\,,\ c_i\,,\ D_{\text{kk}} = V_\tau\right). \tag{6.28}$$

In a general $S^1$ reduction, the fields in the KK photon multiplet would not be linked to those in the gravity multiplet, however in a supersymmetric background satisfying the extra conditions (6.20) this is the case. See Appendix A for more details.

We have thus obtained a supersymmetric 3d background with $U(1) \times U(1)$ symmetry, depending on the three parameters $b, k_1, k_2$. The supersymmetric Killing vector

$$\check{K} = \frac{1}{2\ell} \left[ (b + ik_1) \frac{\partial}{\partial \varphi_1} + (b^{-1} + ik_2) \frac{\partial}{\partial \varphi_2} \right],$$ (6.29)

is generically complex; as such, the background falls out of the analysis of [70].

Specializing to $k_1 = k_2 = 0$ we obtain the elliptically squashed three-sphere of [73]. Taking $b = 1$, $k_1 = \pm k_2 \equiv k$ leads us to more symmetric backgrounds made of a squashed sphere with $SU(2) \times U(1)$ invariance and squashing parameter $\frac{1}{\sqrt{1+k^2}}$. The choice $b = 1$, $k_1 = k_2$ gives the $SU(2) \times U(1)$ invariant background of [73], while the choice $b = 1$, $k_1 = -k_2$ corresponds to the background of [76]. Our background should also be related to (and possibly incorporate) the two-parameter background of [93], which leads to a 3d partition function depending on one complex parameter.

## 6.3 Evaluating the 3d supergravity terms

We now evaluate the supersymmetric contact terms $I_{1,2,3,4}$ listed in Section 2 in the background defined above.

Before coming to that, let us briefly summarize how these contact terms are obtained in 3d new-minimal supergravity. Using the fields in the supergravity multiplet one can define a gauge vector multiplet, dubbed the $R$-symmetry vector multiplet, whose bosonic components are (see e.g. [72])

$$R\text{-symmetry multiplet} = \left( \sigma = H, \ a_i = \check{A}_i - \frac{1}{2} \check{V}_i, \ D = \frac{1}{4} \left( \check{\mathcal{R}} + 2\check{V}_i \check{V}^i + 2H^2 \right) \right).$$ (6.30)

As discussed in [70], from any gauge vector multiplet of 3d new-minimal supergravity with bosinic components $(\sigma, a_i, D)$, one can write down a supersymmetric Chern-Simons action, whose bosonic part reads

$$I_{\text{CS}} = \int_{\mathcal{M}_3} \left( i \, a \wedge da - 2\sigma D \, \text{vol}_3 \right).$$ (6.31)

Applying this to the KK photon multiplet and the $R$-symmetry vector multiplet, we obtain the Chern-Simons terms $I_1, I_2, I_3$ given in (2.4)–(2.6). The $I_4$ term is the $\mathcal{N} = 2$ conformal supergravity action in three dimensions [94].

Evaluating the integrals $I_{1,2,3,4}$ in our background with generic parameters $b, k_1, k_2$ is complicated and requires the aid of a computer, therefore we will just provide the results. For the first three integrals we obtain

$$I_1 = \frac{4\pi^3 i}{\omega_1 \omega_2},$$

$$I_2 = 2\pi^2 \frac{\omega_1 + \omega_2}{\omega_1 \omega_2},$$

$$I_3 = -\frac{\pi i (\omega_1 + \omega_2)^2}{4 \, \omega_1 \omega_2}.$$ (6.32)

These are precisely the expressions expected from the analytic continuation of the result obtained before for real $\omega_1, \omega_2$. From the 3d point of view, it is non-trivial that each of these terms

is a holomorphic function of the parameters (6.15) describing the 4d complex structure. This is however nicely consistent with the fact that we are effectively evaluating a supersymmetric 4d partition function, which is a holomorphic function of such complex structure parameters.

For the gravitational Chern-Simons term we did not succeed in obtaining the expected formula

$$I_4 = \frac{\pi i}{48} \frac{(\omega_1 - \omega_2)^2}{\omega_1 \omega_2} \tag{6.33}$$

in general, however we did obtain it, for instance, for the case where $\omega_1$ is real and $\omega_2$ is complex (with $0 < \operatorname{Re} \omega_2 \leq \omega_1$ so as to satisfy the condition $|p| \leq |q| < 1$). We hope to clarify this puzzling aspect of our analysis in the future. Given that the partition function should be a holomorphic function of $\omega_1, \omega_2$, we continue using (6.33) in spite of the above shortcoming.

It may be useful to summarize our strategy. In the previous sections we analyzed the physics of the KK modes in the small $\beta$ expansion and showed that

$$\log \mathcal{I} = -n_0 \frac{\operatorname{Tr} R^3 - \operatorname{Tr} R}{24} I_1 - \frac{3 \operatorname{Tr} R^3 - \operatorname{Tr} R}{24} I_2 - n_0 \frac{6 \operatorname{Tr} R^3 - \operatorname{Tr} R}{12} I_3 - n_0 \operatorname{Tr} R \, I_4$$
$$+ \log |G| + O(\beta). \tag{6.34}$$

This result holds for $n_0 = \pm 1$. We explicitly derived it for the second sheet $n_0 = +1$, the derivation for $n_0 = -1$ being completely analogous.[21] We then used known expressions for the integrals $I_{1,2,3,4}$ on the direct product background $S^1 \times S_b^3$, with $k_1 = k_2 = 0$, and extended the result to complex values of the chemical potentials by analiticity. In the present section, we have instead explicitly evaluated the contact terms $I_{1,2,3,4}$ for the case where $S_b^3$ is twisted over $S^1$, proving that the analytic extension used before is correct for $I_{1,2,3}$, and in part for $I_4$. Plugging (6.32), (6.33) in (6.34) we get

$$\log \mathcal{I} = \frac{1}{48 \omega_1 \omega_2} \Big[ -8\pi^3 i n_0 (\operatorname{Tr} R^3 - \operatorname{Tr} R) - 4\pi^2 (\omega_1 + \omega_2)(3 \operatorname{Tr} R^3 - \operatorname{Tr} R)$$
$$+ 6\pi i n_0 (\omega_1 + \omega_2)^2 \operatorname{Tr} R^3 - 2\pi i n_0 (\omega_1^2 + \omega_2^2) \operatorname{Tr} R \Big] + \log |G| + O(\beta), \tag{6.35}$$

which specializing to $n_0 = 1$ yields the result in Eq. (1.7).[22]

If the argument given in Section 1 about the contribution of the Casimir energy at $O(\beta)$ and the further subleading terms in the small-$\beta$ expansion is correct, our final result for $n_0 = \pm 1$ can be nicely expressed as

$$\log \mathcal{I} = \frac{(\omega_1 + \omega_2 + 2\pi i n_0)^3}{48 \, \omega_1 \omega_2} \operatorname{Tr} R^3 - \frac{(\omega_1 + \omega_2 + 2\pi i n_0)(\omega_1^2 + \omega_2^2 - 4\pi^2)}{48 \omega_1 \omega_2} \operatorname{Tr} R$$
$$+ \log |G| + O(e^{-\ell/\beta}), \tag{6.36}$$

which for $n_0 = 1$ is the expression given in Eq. (1.8).

As a side remark, we note that our evaluation of $I_2$ also provides a check of the asymptotic formula (1.3) of [20] for the index on the first sheet ($n_0 = 0$) in the case of a general background with $S^1 \times S^3$ topology and two independent complex structure parameters $\omega_1, \omega_2$, that had not been explicitly done so far.

---

[21] One just has to repeat the sum over the KK towers discussed in Sections 3, 4, this time using the Fourier expansion $\chi(\tau) = \sum_{n \in \mathbb{Z}} \chi_n \, e^{\frac{2\pi i}{\beta} \left( n - \frac{R+F}{2} \right) \tau}$.

[22] In order to check agreement of the divergent terms in this formula with [33, 34], one should notice that our parameters $\omega_1, \omega_2$ agree with those in [33]. On the other hand, they are related with the $\sigma, \tau$ parameters used in [34] as $\omega_1 = -2\pi i \sigma$, $\omega_2 = -2\pi i \tau$ (so they differ by a sign from the parameters $\omega_1^{\text{there}}, \omega_2^{\text{there}}$ that appear e.g. in Eq. (2.30) there).

In order to evaluate the integrals $I_{1,2,3,4}$ for the general (possibly twisted) $S^1 \times \mathcal{M}_3$ background, where $\mathcal{M}_3$ is an appropriate Seifert manifold, one just has to put the 4d background in the KK form (6.19) and identify the 3d supergravity fields using the dictionary (6.27). As conjectured at the end of Section 1, this would provide an effective field theory prediction for the asymptotics of the supersymmetric index on the second sheet.

## Acknowledgements

We thank P. Bomans, C. Closset, M. Dedushenko, L. Di Pietro, M. Martone, L. Rastelli, M. Rocek, and S. Razamat for very useful discussions. We thank A. Arabi Ardehali and S. Murthy for communicating some of their preliminary results that led to the publication [95], which appeared on the arXiv on the same day as v1 of this paper. ZK is supported in part by the Simons Foundation grant 488657 (Simons Collaboration on the Non-Perturbative Bootstrap) and the BSF grant no. 2018204.

## A  KK reduction of 4d supergravity variations

In this appendix we show that the circle reduction of the 4d new-minimal supergravity multiplet yields the 3d new-minimal supergravity multiplet together with the KK photon multiplet. We work out the dictionary relating the 4d and 3d multiplets. This revisits Appendix F of [9] and similar reductions in [70, App. D] as well as in [90, Sect. 5]. While these references exploited part of the conditions satisfied in a supersymmetric background to simplify the analysis, here we discuss a general KK reduction of the 4d theory, independently of whether the supersymmetry equations are satisfied or not.

In the main text we used the conditions $e^\Phi = 1$ and $A_\tau = V_\tau$, that were imposed in the reduction discussed in [70, App. D]. Our scope here is to show that these restrictions can in principle be relaxed, without changing the essence of the story.

**3d conventions.** Our Riemannian geometry and 4d spinor conventions are as in [9, App. A], in particular the Ricci scalar of a round sphere is positive. Our 3d conventions are the same as in [9, App. F] and we repeat them here for convenience. We denote by $i, j, k$ the 3d curved indices, and a $\check{\ }$ denotes 3d quantities. For any 3d spinor $\varepsilon$, its Lorentz covariant derivative is defined as

$$\check{\nabla}_i \varepsilon = \left( \partial_i + \frac{i}{4} \check{\omega}_{i\check{a}\check{b}} \epsilon^{\check{a}\check{b}\check{c}} \gamma_{\check{c}} \right) \varepsilon , \tag{A.1}$$

where $\check{\omega}_{i\check{a}\check{b}}$ is the 3d spin connection, and $\check{a}, \check{b}, \check{c} = 1, 2, 3$ are 3d flat indices. Our 3d gamma matrices are identified with the Pauli matrices, $(\gamma^{\check{a}})_\alpha{}^\beta = (\sigma^{\check{a}}_{\text{Pauli}})_\alpha{}^\beta$. These are related to the 4d sigma matrices $\sigma^a_{\alpha\dot\alpha}$, $\tilde{\sigma}^{a\,\dot\alpha\alpha}$, $a = 1, 2, 3, 4$, as

$$\sigma^{\check{a}}_{\alpha\dot\alpha} = i (\gamma^{\check{a}})_\alpha{}^\beta \sigma^4_{\beta\dot\beta} , \qquad \tilde{\sigma}^{\check{a}\,\dot\alpha\alpha} = -i\, \tilde{\sigma}^{4\,\dot\alpha\beta} (\gamma^{\check{a}})_\beta{}^\alpha . \tag{A.2}$$

It follows that

$$\sigma_{\check{a}4} = -\frac{i}{2} \gamma_{\check{a}} , \qquad \sigma_{\check{a}\check{b}} = -\frac{i}{2} \epsilon_{\check{a}\check{b}\check{c}} \gamma^{\check{c}} ,$$

$$\tilde{\sigma}_{\check{a}4} = -\frac{i}{2} \tilde{\sigma}_4 \gamma_{\check{a}} \sigma_4 , \qquad \tilde{\sigma}_{\check{a}\check{b}} = +\frac{i}{2} \epsilon_{\check{a}\check{b}\check{c}} \tilde{\sigma}_4 \gamma^{\check{c}} \sigma_4 . \tag{A.3}$$

A 4d left-handed spinor $\zeta_\alpha$ directly reduces to a 3d spinor, while a 4d right-handed spinor $\tilde{\zeta}^{\dot\alpha}$ is mapped to a 3d spinor via $i\sigma^4_{\alpha\dot\alpha} \tilde{\zeta}^{\dot\alpha}$, or $\tilde{\zeta}_{\dot\alpha} (i\tilde{\sigma}^4)^{\dot\alpha\alpha}$.



**The KK ansatz.** Given a Killing vector $\frac{\partial}{\partial\tau}$, we put the 4d metric in the KK form

$$\mathrm{d}s^2 = \mathrm{e}^{-2\Phi}\breve{g}_{ij}\mathrm{d}x^i\mathrm{d}x^j + \mathrm{e}^{2\Phi}(\mathrm{d}\tau+c)^2\,, \tag{A.4}$$

where we are splitting the 4d coordinates as $x^\mu = (x^i, \tau)$, and $\breve{g}_{ij}$, $c = c_i\mathrm{d}x^i$, $\Phi$ are the 3d metric, the KK photon and the dilaton, depending on the 3d coordinates $x^i$. The Weyl rescaling of the 3d metric ensures that a dimensional reduction of the 4d Einstein-Hilbert term yields a 3d term where the metric is in the Einstein frame. For the form fields we take the same ansatz as in (6.19), that is

$$\mathcal{A}_i = A_i - c_iA_\tau\,, \qquad \mathcal{V}_i = V_i - c_iV_\tau\,. \tag{A.5}$$

The 4d vielbein and its inverse can be chosen as

$$e^a{}_\mu = \begin{pmatrix} \mathrm{e}^{-\Phi}\breve{e}^{\breve{a}}{}_i & 0 \\ \mathrm{e}^{\Phi}c_i & \mathrm{e}^{\Phi} \end{pmatrix}, \qquad e^\mu{}_a = \begin{pmatrix} \mathrm{e}^{\Phi}\breve{e}^i{}_{\breve{a}} & 0 \\ -\mathrm{e}^{\Phi}c_j\breve{e}^j{}_{\breve{a}} & \mathrm{e}^{-\Phi} \end{pmatrix}, \tag{A.6}$$

where $\breve{e}^{\breve{a}}{}_i$ is a vielbein for $\breve{g}_{ij}$, and $\breve{e}^i{}_{\breve{a}}$ is its inverse. The 4d spin connection $\omega_{cab}$ decomposes as

$$\omega_{\breve{c}\breve{a}\breve{b}} = \mathrm{e}^{\Phi}\big(\breve{e}^i{}_{\breve{c}}\,\breve{\omega}_{i\breve{a}\breve{b}} - 2\,\delta_{\breve{c}[\breve{a}}\breve{e}^i{}_{\breve{b}]}\partial_i\Phi\big), \qquad \omega_{4\breve{a}\breve{b}} = -\mathrm{e}^{3\Phi}\partial_{[i}c_{j]}\breve{e}^i{}_{\breve{a}}\breve{e}^j{}_{\breve{b}}\,,$$

$$\omega_{\breve{c}4\breve{b}} = \mathrm{e}^{3\Phi}\partial_{[i}c_{j]}\breve{e}^i{}_{\breve{b}}\breve{e}^j{}_{\breve{c}}\,, \qquad \omega_{44\breve{b}} = \mathrm{e}^{\Phi}\breve{e}^i{}_{\breve{b}}\partial_i\Phi\,. \tag{A.7}$$

**Reduction of the gravitino variation.** We consider new-minimal supergravity [68,69], in its Euclidean version (see e.g. [51]). The gravity multiplet is made of the vielbein $e^a{}_\mu$, the gravitino $\psi_\mu, \widetilde{\psi}_\mu$ and the auxiliary fields $A_\mu, V_\mu$.

We study the reduction along $\frac{\partial}{\partial\tau}$ of the gravitino supersymmetry variation. At the linearized level in the fermion fields this is

$$\delta\psi_\mu = 2\big(\nabla_\mu - iA_\mu + iV_\mu + iV^\nu\sigma_{\mu\nu}\big)\zeta\,, \tag{A.8}$$

$$\delta\widetilde{\psi}_\mu = 2\big(\nabla_\mu + iA_\mu - iV_\mu - iV^\nu\widetilde{\sigma}_{\mu\nu}\big)\widetilde{\zeta}\,. \tag{A.9}$$

We assume that $\psi_\mu, \widetilde{\psi}_\mu, \zeta, \widetilde{\zeta}$ are independent of $\tau$. Importantly, this condition is satisfied by the boundary conditions considered in the main text. Reducing $\delta\psi_\mu$ we obtain the following 3d variations

$$\delta(\psi_i - c_i\psi_\tau) = \Big[2(\breve{\nabla}_i - i\mathcal{A}_i + i\mathcal{V}_i) + \epsilon_{ijk}\big(-i\partial^j\Phi + \tfrac{1}{2}\mathrm{e}^{2\Phi}v^j + \mathcal{V}^j\big)\gamma^k + \mathrm{e}^{-2\Phi}V_\tau\gamma_i\Big]\zeta\,, \tag{A.10}$$

$$\delta\psi_\tau = \Big[\mathrm{e}^{2\Phi}\big(\tfrac{1}{2}\mathrm{e}^{2\Phi}v_i - i\,\partial_i\Phi - \mathcal{V}_i\big)\gamma^i - 2i\,(A_\tau - V_\tau)\Big]\zeta\,, \tag{A.11}$$

where we introduced

$$v^i = -i\,\epsilon^{ijk}\partial_j c_k\,. \tag{A.12}$$

The 3d indices $i, j$ are always lowered/raised using the 3d metric $\breve{g}_{ij}$ and its inverse $\breve{g}^{ij}$. The reduction of $\delta\widetilde{\psi}_\mu$ works in a similar way and yields

$$i\sigma_4\delta(\widetilde{\psi}_i - c_i\widetilde{\psi}_\tau) = \Big[2(\breve{\nabla}_i + i\mathcal{A}_i - i\mathcal{V}_i) + \epsilon_{ijk}\big(-i\partial^j\Phi - \tfrac{1}{2}\mathrm{e}^{2\Phi}v^j - \mathcal{V}^j\big)\gamma^k + \mathrm{e}^{-2\Phi}V_\tau\gamma_i\Big]i\sigma_4\widetilde{\zeta}\,, \tag{A.13}$$

$$i\sigma_4\,\delta\widetilde{\psi}_\tau = \Big[\mathrm{e}^{2\Phi}\big(\tfrac{1}{2}\mathrm{e}^{2\Phi}v_i + i\,\partial_i\Phi - \mathcal{V}_i\big)\gamma^i + 2i\,(A_\tau - V_\tau)\Big]i\sigma_4\widetilde{\zeta}\,. \tag{A.14}$$

**Identification of the 3d supergravity fields.** We want to interpret the variations above as supersymmetry variations in three-dimensional new-minimal supergravity. The new-minimal supergravity multiplet is

$$\text{supergravity multiplet} = \left( \check{g}_{ij} \,,\, \check{\psi}_i \,,\, \widetilde{\check{\psi}}_i \,,\, \check{A}_i \,,\, \check{V}_i \,,\, H \right), \tag{A.15}$$

while the KK photon vector multiplet is

$$\text{KK photon multiplet} = \left( \varsigma_{\text{kk}} \,,\, c_i \,,\, \lambda_{\text{kk}} \,,\, \widetilde{\lambda}_{\text{kk}} \,,\, D_{\text{kk}} \right). \tag{A.16}$$

We identify the 3d gravitino and gaugino as

$$\check{\psi}_i = \mathrm{e}^{\frac{\Phi}{2}} \left( \psi_i - c_i \psi_\tau + i\,\mathrm{e}^{-2\Phi} \gamma_i \psi_\tau \right), \qquad \widetilde{\check{\psi}}_i = \mathrm{e}^{\frac{\Phi}{2}} i\sigma_4 (\widetilde{\psi}_i - c_i \widetilde{\psi}_\tau - i\,\mathrm{e}^{-2\Phi} \gamma_i \widetilde{\psi}_\tau), \tag{A.17}$$

$$\lambda_{\text{kk}} = 2\,\mathrm{e}^{-\frac{7}{2}\Phi} \psi_\tau, \qquad \widetilde{\lambda}_{\text{kk}} = 2\,\mathrm{e}^{-\frac{7}{2}\Phi} i\sigma_4 \widetilde{\psi}_\tau, \tag{A.18}$$

while the 3d $\mathcal{N}=2$ spinor parameters are given by

$$\varepsilon = \mathrm{e}^{\frac{\Phi}{2}} \zeta, \qquad \widetilde{\varepsilon} = \mathrm{e}^{\frac{\Phi}{2}} i\sigma_4 \widetilde{\zeta}. \tag{A.19}$$

The bosonic fields in the three-dimensional supergravity multiplet are identified as

$$\check{V}_i = 2\mathcal{V}_i, \qquad \check{A}_i = \mathcal{A}_i + \frac{3}{2}\mathcal{V}_i + \frac{i}{4}\mathrm{e}^{2\Phi}\epsilon_{ijk}\partial^j c^k, \qquad H = \mathrm{e}^{-2\Phi}(2A_\tau - V_\tau), \tag{A.20}$$

while the bosonic fields in the KK photon vector multiplet besides $c_i$ itself are given by

$$\varsigma_{\text{kk}} = -\mathrm{e}^{-2\Phi}, \qquad D_{\text{kk}} = -\mathrm{e}^{-4\Phi}(2A_\tau - 3V_\tau). \tag{A.21}$$

One can check that

$$\check{\nabla}_i \check{V}^i = 0, \tag{A.22}$$

where $\check{\nabla}$ is the Levi-Civita connection of the 3d metric $\mathrm{d}\check{s}^2$. This allows to identify $\check{V}^i$ as the one-form dual to the a gauge field strength, as required by 3d new-minimal supergravity.

Using these identifications, the 3d gravitino variations take the form

$$\delta\check{\psi}_i = 2\left( \check{\nabla}_i - i\check{A}_i + i\check{V}_i \right)\varepsilon + H\gamma_i\varepsilon + \epsilon_{ijk}\check{V}^j\gamma^k\varepsilon, \tag{A.23}$$

$$\delta\widetilde{\check{\psi}}_i = 2\left( \check{\nabla}_i + i\check{A}_i - i\check{V}_i \right)\widetilde{\varepsilon} + H\gamma_i\widetilde{\varepsilon} - \epsilon_{ijk}\check{V}^j\gamma^k\widetilde{\varepsilon}, \tag{A.24}$$

while the 3d gaugino variation reads

$$\delta\lambda_{\text{kk}} = \left( -i\epsilon_{ijk}\partial^j c^k - i\partial_i\varsigma_{\text{kk}} + \varsigma_{\text{kk}}\check{V}_i \right)\gamma^i\varepsilon + i(D_{\text{kk}} + \varsigma_{\text{kk}}H)\varepsilon,$$

$$\delta\widetilde{\lambda}_{\text{kk}} = \left( -i\epsilon_{ijk}\partial^j c^k + i\partial_i\varsigma_{\text{kk}} + \varsigma_{\text{kk}}\check{V}_i \right)\gamma^i\widetilde{\varepsilon} - i(D_{\text{kk}} + \varsigma_{\text{kk}}H)\widetilde{\varepsilon}. \tag{A.25}$$

These match the fermionic variations in three-dimensional new-minimal supergravity at the linear level in the fermions [70].

If we impose $\psi_\tau = \widetilde{\psi}_\tau = 0$ together with $\delta\psi_\tau = 0$, $\delta\widetilde{\psi}_\tau = 0$, corresponding to part of the supersymmetry conditions for a bosonic background, and in addition require $\mathrm{e}^\Phi = 1$, $A_\tau = V_\tau$, then from (A.11), (A.14) we infer that $\mathcal{V}_i = -\frac{i}{2}\epsilon_{ijk}\partial^j c^k$. It follows that the identifications (A.20) for the fields in the 3d new-minimal supergravity multiplet reduce to those given in Appendix D of [70], that we reported in (6.27). Using the present more general identifications, we could have avoided imposing $\mathrm{e}^\Phi = 1$ and $A_\tau = V_\tau$ in Section 6.2, and thus we could have avoided fixing the 4d conformal factor $\Omega$ and the function $\kappa$ as specified there. For instance, we could have taken $\Omega = 1$ and $\kappa = 0$ instead. We have performed a preliminary evaluation of the supersymmetric integrals $I_{1,2,3,4}$ using this alternative choice with some restricted choice of the parameters $b, k_1, k_2$, and, at least for this restricted choice, we have obtained the same results.

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
