# Peer review of "EFT and the SUSY Index on the 2nd Sheet"

_SciPost Physics, doi:SciPost Phys. 11, 004 (2021)_

## Round 2 · Referee Report · Anonymous (Referee 1) · 2021-5-24

Strengths

1- The introduction is a beautifully written, clear description of the general subject and the results of the present paper. There is an enormous body of past literature in the subject of computing BPS operator counting indices for 4d SCFTs, much of it quite technical. Past literature noticed, but did not fully explain, many intriguing aspects of the mathematical results, including analytically continuing in the chemical potentials and connecting to 3d theories. The introduction does a beautiful job of summarizing these results, and provides clear physical explanations for when and why many of the past observations are true. The introduction should be required reading for anyone working on supersymmetric partition functions.

2- sections 2,3,4 go into the more technical details, but are still beautifully clear. Section 2 is on general aspects of the 3d effective theory coming from putting the 4d theory on a circle, including the background supergravity needed for the study of the 3d effective theory on a compact space. Sections 3 and 4 discuss the associated contributions to the partition function.

3- section 5 gives a pedagogical analysis of the most interesting examples. In particular, it illustrates how some theories have only one sheet, pointing out and resolving an apparent puzzle with the general remarks from the earlier sections.

4- section 6 generalizes the results to general twisted backgrounds. Again, everything is worked out and written up in a very clear and useful way.

5- the appendix contains useful technical details about the KK reduction of 4d SUGRA variations.

Weaknesses

I do not see this paper as having any notable weaknesses. It contains interesting new results, and is exceptionally clearly written.

Report

I enthusiastically recommend publishing this paper. It synthesizes and explains many past observations from the literature, and clearly shows when and why many past conjectural results are valid. The results here will certainly be useful for the research community, and I anticipate that it will have a high impact.

Requested changes

none

---

## Round 2 · Referee Report · Anonymous (Referee 2) · 2021-7-2

Report

The main result of this manuscript is summarized in equation (4.4), i.e. a prediction for an exact expansion of the four-dimensional superconformal index (of a large class of 4d SCFTs) in a specific $``$large temperature$''$ expansion (called $\beta\to 0$ limit in the second sheet).

By using a 3d EFT approach, the authors compute the O$(\frac{1}{\beta^2})$, O$(\frac{1}{\beta})$, and part of the O$(\beta^0)$ contribution reported in (4.4). Under an assumption, the remaining O($\beta^0$) contribution is predicted to count the number of degenerate vacua emerging after the spontaneous breaking of a one-form symmetry, provided such symmetry exists in the UV theory. (4.4) is conjectured to be exact up to exponentially small contributions.

Some of these results have been previously reported. For example, the terms of order O($\frac{1}{\beta^2}$) and~O$(\frac{1}{\beta})$ have been explicitly computed before (for instance, in references [31,32,33,34]). A proposal for the contributions of order O($\beta^0$) has also been previously reported (in reference [37]). The prediction of this manuscript matches the proposal of [37], but it does it from a completely independent and original perspective.

Although a complete computation of the O($\beta^0$) term is still an open problem, the form conjectured in (4.4) is expected to be correct. The 3d EFT theory approach and the physical arguments developed in this manuscript offer a fresh perspective that will be useful for future results.

The paper is very well-written. The explanations are very clear, and the used assumptions are clearly stated.

---

## Editorial Decision

published